# Improved Coresets for Vertical Federated Learning: Regularized Linear and Logistic Regressions

**Supratim Shit** [1]  **Gurmehak Kaur Chadha** [1]  **Surendra Kumar** [1]  **Bapi Chatterjee** [1]

## Abstract

Coreset, as a summary of training data, offers an efficient approach for reducing data processing and storage complexity during training. In the emerging vertical federated learning (VFL) setting, where scattered clients store different data features, it directly reduces communication complexity. In this work, we introduce coresets construction for regularized logistic regression both in centralized and VFL settings. Additionally, we improve the coreset size for regularized linear regression in the VFL setting. We also eliminate the dependency of the coreset size on a property of the data due to the VFL setting. The improvement in the coreset sizes is due to our novel coreset construction algorithms that capture the reduced model complexity due to the added regularization and its subsequent analysis. In experiments, we provide extensive empirical evaluation that backs our theoretical claims. We also report the performance of our coresets by comparing the models trained on the complete data and on the coreset.

## 1. Introduction

Let $\mathbf{Z}$ be a set of $n$ points and their corresponding labels/responses. Here, $\mathbf{Z}$ consists of $\mathbf{X} \in \mathbb{R}^{n \times d}$ represents the $n$ points in $\mathbb{R}^d$ space and labels $\mathbf{y} \in \mathbb{R}^n$. Let $\mathbf{z}_i$ represents the $i^{th}$ point (i.e., $\mathbf{x}_i$) and its corresponding label (i.e., $y_i$) (see; Section 2 for notation). Let $\mathcal{Q}$ be the set of models on which a machine learning algorithm optimizes its loss function. Let, the algorithm uses a nonnegative function $f : \mathbf{Z} \times \mathcal{Q} \to \mathbb{R}_{\geq 0}$ to compute the loss on the dataset for a given model $\mathbf{q} \in \mathcal{Q}$ as, $\text{loss}(\mathbf{Z}, \mathbf{q}) = \sum_{i \in [n]} f(\mathbf{z}_i, \mathbf{q})$.

Regularization is a common technique to control model com-

[1]Department of Computer Science and Engineering, Indraprastha Institute of Information Technology Delhi, New Delhi, India. Correspondence to: Supratim Shit <supratim@iiitd.ac.in>, Bapi Chatterjee <bapi@iiitd.ac.in>.

*Proceedings of the $42^{nd}$ International Conference on Machine Learning*, Vancouver, Canada. PMLR 267, 2025. Copyright 2025 by the author(s).

plexity and to avoid overfitting during training. If the model is regularized by a user-defined parameter $\lambda \in \mathbb{R}_{>0}$, then the loss function is also penalized by $\lambda$ and the model. Thus, it is defined as $\text{loss}(\mathbf{Z}, \mathbf{q}, \lambda) = \sum_{i \in [n]} f(\mathbf{z}_i, \mathbf{q}) + g(\lambda, \mathbf{q})$, where $g(\cdot)$ adds a regularization penalty to the unregularized loss function. In this paper, we focus on regularized logistic regression and ridge regression problems. For the above defined dataset $\mathbf{Z}$, regularization parameter $\lambda > 0$ and a set of models $\mathcal{Q}$, the losses of regularized logistic regression and ridge regression for any model $\mathbf{q} \in \mathcal{Q}$ are defined as,

$$
\begin{aligned}
\mathsf{ClassLoss}(\mathbf{Z}, \mathbf{q}, \lambda) \quad := \quad & \sum_{i=1}^{n} \ln\left(1 + \exp(-y_i \mathbf{x}_i^\top \mathbf{q})\right) \\
& + \lambda \|\mathbf{q}\|_1 \quad\quad (1) \\
\mathsf{RegLoss}(\mathbf{Z}, \mathbf{q}, \lambda) \quad := \quad & \sum_{i=1}^{n} (\mathbf{x}_i^\top \mathbf{q} - y_i)^2 + \lambda \|\mathbf{q}\|_2^2 \quad (2)
\end{aligned}
$$

In the vertical federated learning (VFL) setting, there are multiple scattered clients, so no clients have access to the complete feature space. In the VFL setting, we study the setup where the feature space is partitioned between clients. Formally, let there are $T$ scattered clients, we consider that a dataset $\mathbf{Z}$ is partitioned among all the clients, each having $\{\mathbf{Z}^{(1)}, \mathbf{Z}^{(2)}, \ldots, \mathbf{Z}^{(T)}\}$, such that no two clients share any features and their union is $\mathbf{Z}$. In this work, we present coreset construction algorithms for the following two crucial machine learning problems in the VFL model.

**Definition 1.1** (**V**ertical **R**egularized **Log**istic Regression **(VRLog)**). Given a dataset $\mathbf{Z}$ consisting of $\mathbf{X}$ representing the points and $\mathbf{y}$ be their labels in the VFL model, a regularization parameter $\lambda > 0$, the goal of the VRLog problem is to compute a vector $\mathbf{q} \in \mathbb{R}^d$ on the server that (approximately) minimizes $\mathsf{ClassLoss}(\mathbf{Z}, \mathbf{q}, \lambda)$ while maintaining minimum total communication complexity.

**Definition 1.2** (**V**ertical **R**idge **L**inear **R**egression **(VRLR)**). Given a dataset $\mathbf{Z}$ consisting of $\mathbf{X}$ representing the points and $\mathbf{y}$ be their responses in the VFL model, regularization parameter $\lambda > 0$, the goal of the VRLR problem is to compute a vector $\mathbf{q} \in \mathbb{R}^d$ on the server that (approximately) minimizes $\mathsf{RegLoss}(\mathbf{Z}, \mathbf{q}, \lambda)$ while maintaining minimum total communication complexity.

For training a model in a VFL setting, the communication cost grows proportionately to the data size, thus entails finding approaches of reducing the training data without compromising the trained model quality. So, to address this issue, we take advantage of coresets. At a high level, a coreset is a small summary of the original dataset that approximates the learning objective for every possible choice of learning parameters. For VRLog and VRLR, we give algorithms that return a weighted subset ensuring the following guarantees.

**Definition 1.3.** Let $\mathbf{Z}$ be a dataset as described in the Definition 1.1 in the VFL setting. Let $\varepsilon \in (0, 1)$, $\lambda > 0$. Let $\mathbf{S}_w$ be a weighted set, comprising of a subset $\mathbf{S} \subseteq \mathbf{Z}$ with an associated weight function $w : \mathbf{S} \to [1, \infty)$. We call $\mathbf{S}_w$ an $\varepsilon$-coreset for VRLog if with at least 0.99 probability, it guarantees that for every $\mathbf{q} \in \mathbb{R}^d$.

$$\mathsf{ClassLoss}(\mathbf{S}_w, \mathbf{q}, \lambda) \in (1 \pm \varepsilon) \cdot \mathsf{ClassLoss}(\mathbf{Z}, \mathbf{q}, \lambda),$$

where loss on $\mathbf{S}_w$ is $\mathsf{ClassLoss}(\mathbf{S}_w, \mathbf{q}, \lambda)$ defined as $\sum_{i \in [|\mathbf{S}|]} w(i) \cdot \ln(1 + \exp(-y_i \cdot \mathbf{x}_i^\top \mathbf{q})) + \lambda \|\mathbf{q}\|_1$.

**Definition 1.4.** Let $\mathbf{Z}$ be a dataset as described in the Definition 1.2 in the VFL setting. Let $\varepsilon \in (0, 1)$, $\lambda > 0$. Let $\mathbf{S}_w$ be a weighted set, comprising of a subset $\mathbf{S} \subseteq \mathbf{Z}$ with an associated weight function $w : \mathbf{S} \to [1, \infty)$. We call $\mathbf{S}_w$ an $\varepsilon$-coreset for VRLog if with at least 0.99 probability, it guarantees that for every $\mathbf{q} \in \mathbb{R}^d$.

$$\mathsf{RegLoss}(\mathbf{S}_w, \mathbf{q}, \lambda) \in (1 \pm \varepsilon) \cdot \mathsf{RegLoss}(\mathbf{Z}, \mathbf{q}, \lambda),$$

where $\mathsf{RegLoss}(\mathbf{S}_w, \mathbf{q}, \lambda) := \sum_{i \in [|\mathbf{S}|]} w(i)(\mathbf{x}_i^\top \mathbf{q} - y_i)^2 + \lambda \|\mathbf{q}\|_2^2$.

It is important to note that even though we consider a VFL setup, where the features of the dataset have been partitioned among multiple clients, however the ensured coreset guarantees are on the loss functions defined on the loss functions defined on the complete features of the dataset. Thin inherently possesses some immediate challenges, which we discuss later.

A unified algorithm for constructing a coreset in VFL setting was introduced in (Huang et al., 2022). For completeness, we restate it as algorithm 3. It uses importance sampling for constructing a coreset by computing local importance scores at each client for every point. In this paper, we focus on constructing a coreset for VRLog and VRLR. Next, we discuss our main contributions in this paper.

- We propose a novel algorithm for constructing coresets for centralized regularized logistic regression (see Theorem 5.6). For the VRLog problem, we employ algorithm 1 to locally compute the importance scores for every point, at each client using $\ell_1$ Lewis weights. The computed scores are then served as input to the

algorithm 3. Particularly, in the VFL setup, we show how to aggregate the locally computed scores so that it is sufficient to ensure a global guarantee. One of the crucial contributions here is that for both cases, our algorithm effectively captures the reduction in the model complexity due to regularization. We analyze and show that for $\lambda > 0$, the algorithm returns a coreset, whose size decreases with an increasing $\lambda$ (See; Corollary 5.7).

- For the VRLR problem, we propose the algorithm 2 to compute the local importance scores for every point at each client. We further show that when these scores are used as input for algorithm 3, the resulting coreset has a size that decreases as the regularization parameter $\lambda$ increases (See Theorem 6.1).

- Intuitively, regularization reduces the model complexity. The model complexity decreases with an increasing regularization parameter, $\lambda$. The size of the coresets returned from algorithm 3 complements this phenomenon. This is due to the importance scores returned by both algorithms 1 and 2 for VRLog and VRLR, respectively. Both algorithm incorporates the regularization penalty to the original partitioned dataset at each client. This dilutes each point's sensitivity (see Definition 4.1), which in turn lowers its importance score. As the coreset size depends on the total sensitivity, we meticulously analyze this and show that it is equal to $\ell_1$ and $\ell_2$ *statistical dimension* (see Definition 5.4) of the data with respect to a regularization $\lambda$ for VRLog and VRLR respectively.

- Finally, we performed an extensive empirical evaluation for both problems in the VFL setup. Our experiments not only support our theoretical guarantees but also show that our algorithm outperforms other coreset construction algorithms in the same setup. We compare the performance in multiple metrics on both training and test data. We also show that the model trained on our coresets is close to the model trained on the full training dataset.

## 2. Model and Preliminaries

**Notations:** A scalar is denoted by a lowercase letter, e.g., $p$ while a vector is denoted by a boldface lowercase letter, e.g., $\mathbf{x}$. By default, all vectors are considered as column vectors unless specified otherwise. Matrices or sets are denoted by boldface uppercase letters, e.g., $\mathbf{X}$. Specifically, $\mathbf{X}$ denotes an $n \times d$ matrix where $n$ is the number of points (or rows) and the feature space is $\mathbb{R}^d$. Normally, $\mathbf{x}_i^\top$ and $\mathbf{x}_j$ represents the $i^{th}$ row and $j^{th}$ column respectively of the matrix $\mathbf{X}$, unless stated otherwise. We consider the case where $n \gg d$. For any $p \in [1, 2]$ we denote $\ell_p$ norm for a vector $\mathbf{x}$ as $\|\mathbf{x}\|_p =$

$(\sum_i x_i^p)^{1/p}$. The square of the Frobenius norm of a matrix is defined as $\|\mathbf{X}\|_F^2 := \sum_{i,j} x_{i,j}^2$. For any $p \in (0, \infty)$, except when $p = 2$, the $\ell_p$ norm of a matrix is defined as $\|\mathbf{X}\|_p := \left(\sum_{i,j} x_{i,j}^p\right)^{1/p}$. In this paper, by default, regularized linear regression would mean ridge regression. Let $a$ and $b$ be two scalars such that, $(1 - \varepsilon)a \leq b \leq (1 + \varepsilon)a$. We represent this relation by $b \in (1 \pm \varepsilon)a$. In asymptotic terms, such as coreset size, we use $\tilde{O}(\cdot)$ to hide logarithmic terms.

## 2.1. Coresets

Coresets are weighted samples of datasets (Feldman & Langberg, 2011) with provable theoretical guarantees. In general, the size of the coreset depends on the optimization function (i.e., its model complexity) and the size of the feature space. For a given weighted dataset $\mathbf{Z}$ with an associated weight function $\upsilon : \mathbf{Z} \to \mathbb{R}_{>0}$, the goal of a machine learning algorithm is to optimize a loss function that uses a function $f : \mathbf{Z} \times \upsilon \times \mathbf{Q} \to \mathbb{R}_{\geq 0}$. Here $\mathbf{Q}$ is the set of feasible model parameters. Then for a parameter $\varepsilon \in (0, 1)$ and $\delta \in (0, 1)$ a subset $\mathbf{S} \subset \mathbf{Z}$ with a weight function $w : \mathbf{S} \to \mathbb{R}_{>0}$ is called an $(\varepsilon, \delta)$-coreset if it satisfies the following with at least $1 - \delta$ probability for every $\mathbf{q} \in \mathbf{Q}$.

$$(1 - \varepsilon)f(\mathbf{Z}_\upsilon, \mathbf{q}) \leq f(\mathbf{S}_w, \mathbf{q}) \leq (1 + \varepsilon)f(\mathbf{Z}_\upsilon, \mathbf{q}). \quad (3)$$

For simplicity, we denote the weighted sets as $\mathbf{Z}_\upsilon$ and $\mathbf{S}_w$. Our result holds for any arbitrary weight function $\upsilon : \mathbf{Z} \to \mathbb{R}_{>0}$ and $\delta \in (0, 1)$. However, for simplicity in this paper, we assume $\upsilon : \mathbf{Z} \to 1$ and $\delta = 0.01$ for simplicity. Consequently, our coresets are $\varepsilon$-coresets, which ensures the above guarantees with at least 0.99 probability.

For both problems, our coreset construction algorithm relies on an importance-based sampling method. Every point in the dataset gets a score that intuitively captures the importance or relevance of the point during the training phase. Points are sampled based on these scores, i.e., a point with a higher score will have a higher chance of getting sampled. Next, every sampled point uses these scores to define its weight, eventually reflecting its significance during the training. Finally, the set of weighted sampled points guarantees (3). We have used a standard coreset construction framework (Feldman & Langberg, 2011; Chhaya et al., 2020a) that comprises the following steps.

1. **Importance Score:** For a given dataset and an optimization function, we define a function (aka *sensitivity function*) that captures the importance of every point with respect to the complete dataset.

2. **Distribution:** Next, we derive a tight upper bound for these functions and define a distribution.

3. **Weighted Sample:** Sample points based on the distribution and assign weights inversely proportional to the

sampling probability and the coreset size.

4. **Coreset Guarantee:** Compute the sum of the upper bounds and the VC dimension of the model. Based on these, sampling enough points ensures the desired coreset guarantee.

The main idea is to ensure that the returned weighted subsample is an unbiased estimator with a limited variance.

## 2.2. Federated Learning

Federated learning has become a go-to approach for training machine learning models on a distributed system of clients where communicating data is precluded (Kairouz et al., 2021). Often, the distributed system includes a designated node, called *server*, that stores a synchronized state of the model being trained over peer nodes or *clients*. The server orchestrates the client selection and synchronization methodology. Federated Learning comes in two flavours: (a) Horizontal Federated learning (HFL), where data with entire feature space is available on individual clients; data remains client local and can not be shared with either the server or a peer client, (b) Vertical Federated Learning (VFL), where data is distributed among clients in such a way that they contain a subset of feature space. More formally,

1. **HFL:** Consider a model $\mathbf{q}$ and a set of clients $T$. A basic federated learning procedure *Federated Averaging* (McMahan et al., 2017) is described as

$$\mathbf{q}_{r,k+1}^{(j)} = \mathbf{q}_{r,k}^{(j)} - \eta \nabla_{\mathbf{q}^{(j)}} \mathsf{cost}(\mathbf{Z}^{(j)}, \mathbf{q}_{r,k}^{(j)}) \quad (4)$$

$$\forall j \in \mathbf{S}_r \subseteq [T], \forall k \in [K - 1], \ \forall r \in [R]$$

$$\mathbf{q}_{r+1} = \frac{1}{|\mathbf{S}_r|} \sum_{j \in \mathbf{S}_r} \mathbf{q}_{r,K}^{(j)}, \quad (5)$$

where at each synchronization round $r \in [R]$, $\mathbf{S}_r \subseteq [T]$ clients participate in local training for $K - 1$ steps following (4). $\eta > 0$ is the learning rate, which we take as a constant for simplicity. They synchronize at the server by averaging the local models as in (5). The clients $j \in [T]$ store local data $\mathbf{Z}^{(j)}$. The server sends the synchronized state back to a new subset of clients at every synchronization round $r \in [R]$.

Horizontal federated learning suffers from heterogeneity in data distribution and participation frequency across clients. To address the issues, several improvements have appeared in the literature: FedProx (Li et al., 2020), SCAFFOLD (Karimireddy et al., 2020), Adaptive Federated Optimization (Reddi et al., 2020) are some of the well-known methods.

2. **VFL:** Here over a set of clients $T$ we consider the partition of feature space of data. We denote the dataset

with subset of features partitioned over client set $[T]$ as $\mathbf{X}^{(j)}$ such that $\cup_{j \in [T]} \mathbf{X}^{(j)} = \mathbf{X}$. Thereby, a basic VFL scheme can be described in (6).

$$\text{Client } j \in [T] \text{ computes } \nabla_{\mathbf{q}^{(j)}} \text{cost}(\mathbf{X}^{(j)}, \mathbf{q}_r^{(j)}).$$

$$\mathbf{q}_{r+1} = \mathbf{q}_r - \eta \bigcup_{j \in [T]} \nabla_{\mathbf{q}^{(j)}} \text{cost}(\mathbf{X}^{(j)}, \mathbf{q}_r^{(j)})$$

$$\forall r \in [R], \quad \forall j \in [T]. \quad (6)$$

In VFL setting the server orchestrates the accumulation of gradients computed at the clients before performing a step of gradient descent. The cost of communication is very high as the server has to wait for gradient accumulation and therein a perfect synchronization. Furthermore, each client has to participate in the process, and one step of the gradient update includes a full pass over each client. An early work on VFL appeared in (Hardy et al., 2017). A recent survey on VFL can be found in (Liu et al., 2024).

## 3. Related Work

Coresets have been extensively studied for numerous machine learning models, ranging from clustering (Feldman & Langberg, 2011; Cohen-Addad et al., 2021; 2022; Shit et al., 2022; Chhaya et al., 2022), regression (Avron et al., 2017; Chhaya et al., 2020a), classification (Mai et al., 2021; Tukan et al., 2022) to deep neural networks (Mirzasoleiman et al., 2020; Maalouf et al., 2022). The regularized machine learning models are common in practice, but to the best of our knowledge, the study of their coreset construction algorithms is limited to a few models (Avron et al., 2017; Chhaya et al., 2020b; Ranjan & Shit, 2024). In this work, we introduce a coreset construction algorithm for regularized logistic regression in both centralized and VFL setups. We also improve the coreset size for regularized regression, but in a VFL setup. Lewis weights (Lewis, 1978) are used for coreset construction in a centralized setup where preserving $\ell_p$ subspace is important for real value of $p$ (Cohen & Peng, 2015; Fazel et al., 2022).

Today, the literature on federated learning is sufficiently mature with ever-improving developments. A comprehensive report on the promises of this framework appeared in (Kairouz et al., 2021). Improving federated learning via coreset construction has attracted only limited attention from the research community. (Sivasubramanian et al., 2024) presented a horizontal federated learning method where, at every synchronization round, the gradients are computed based on a coreset of local data that uses submodular functions, which are not tractable. Given the limited size of data available on a large number of clients in the majority of horizontal federated learning applications, the impact of such a construction is potentially limited.

The closest to our work is (Huang et al., 2022). They presented a framework for coreset construction for regularized linear regression and k-means clustering in the VFL setting. As discussed, the complexity of VFL is directly related to the dataset, which has the same cardinality across clients. Clearly, constructing coresets directly reduces data processing and benefits the communication overhead. Compared to (Huang et al., 2022), our work improves by (1) giving a *new coreset construction* for regularized logistic regression, (2) ensuring the *coreset size for ridge regression is optimum*.

## 4. Coreset Construction in VFL

We first state our VFL setup. Consider the dataset $\mathbf{Z}$, consisting of $\mathbf{X}$ representing $n$ points in $\mathbb{R}^d$ and $\mathbf{y} \in \mathbb{R}^n$ representing their labels. We have $[T]$ scattered clients such that every client has only limited access to the feature space, and no two clients share any features. A client $j \in [T]$ has access to all the points but only a limited number of features, which is represented by $\mathbf{Z}^{(j)}$. Now we describe the datasets for both problems in detail.

**VRLog:** $\mathbf{Z} \in \mathbb{R}^{n \times d}$ be the datset where, $\mathbf{z}_i = -y_i \cdot \mathbf{x}_i \in \mathbb{R}^d$ for every $i \in [n]$. Hence, for every client $j \in [T]$, $\mathbf{Z}^{(j)} \in \mathbb{R}^{n \times d_j}$, where $\mathbf{z}_i^{(j)} = -y_i \cdot \mathbf{x}_i^{(j)}$ for every $i \in [n]$. Here, $\sum_{i=1}^{T} d_i = d$. Let $\lambda > 0$ be the regularization parameter.

**VRLR:** $\mathbf{Z} \in \mathbb{R}^{n \times d+1}$ be the dataset where $\mathbf{z}_i = [\mathbf{x}_i, y_i] \in \mathbb{R}^{d+1}$ for every $i \in [n]$,. Hence, for every client $j \in [T-1]$, $\mathbf{Z}^{(j)} \in \mathbb{R}^{n \times d_j}$, where $\mathbf{z}_i^{(j)} = \mathbf{x}_i^{(j)}$ for every $i \in [n]$. For the client $T$, $\mathbf{Z}^{(T)} \in \mathbb{R}^{n \times d_T}$ where $\mathbf{z}_i^{(T)} = [\mathbf{x}_i^{(T)}, y_i]$ for every $i \in [n]$. Here, $\sum_{i=1}^{T} d_i = d + 1$. Let $\lambda > 0$ be the regularization parameter.

For both problems, we use the sensitivity framework for importance sampling, which relies on importance scores (sensitivity scores) of every point. Key challenges in this framework are obtaining a tight upper bound on the sensitivity scores and bounding the total sensitivity. Getting a tighter upper bound on sensitivity scores is often as expensive as solving the actual problem (Braverman et al., 2021). Here, we show how coresets can significantly reduce communication overhead while training a model in VFL. We accomplished this by addressing the following challenges.

**P1:** Even in the VFL setup, our coreset guarantees hold for the global model, similar to a centralized setting where standard sensitivity scores are well-defined. These bounds are typically derived by functions that have access to the complete feature space. However, in a VFL setup, where clients only possess partial feature sets, determining a tight upper bound on the sensitivity score is unknown.

**P2:** The model complexity of a machine learning algorithm

reduces due to an added regularization. As a result, it is natural to expect a smaller coreset size for this problem compared to an unregularized version of the problem. However, designing an algorithm that captures the reduced model complexity for a general problem through the sensitivity scores and then using them to quantify the size of the final coreset is unknown.

We first introduce the sensitivity scores that capture the importance of a point under the reduced model complexity in a centralized setting. In our definition, regularization inherently reduces the importance of each point. As the regularization parameter increases, sensitivity scores decrease accordingly. This is intuitively correct as higher regularization leads to smaller model weights (or norm). The optimal model tends to a zero vector for a very large regularization parameter $\lambda$. In such a case, the sensitivity scores would also be close to 0 (i.e., negligible importance of every point).

**Definition 4.1** (**Regularized Sensitivity**). Let $\mathbf{Z}$ be a dataset with $n$ points along with its labels. Let $\mathcal{Q}$ be the feasible model space and $\lambda \in \mathbb{R}_{>0}$ be a regularization parameter. Let, $\text{loss}(\mathbf{Z}, \mathbf{q}, \lambda) = \sum_{i=1}^{n} f(\mathbf{z}_i, \mathbf{q}) + g(\lambda, \mathbf{q})$ for every $\mathbf{q} \in \mathcal{Q}$. Then for every point $i \in [n]$ we define the regularized sensitivity score as,

$$s_i := \sup_{\mathbf{q} \in \mathcal{Q}} \frac{f(\mathbf{z}_i, \mathbf{q})}{\text{loss}(\mathbf{Z}, \mathbf{q}, \lambda)}$$

In the above definition, the importance of every point $i \in [n]$ is quantified by $s_i$, which is the supremum of the relative loss of the point to the complete regularized loss over all feasible models. Notice that the sensitivity scores can be any value between 0 and 1. Further, as $\lambda$ increases, the sensitivity score decreases. Hence, compared with an unregularized machine learning model for any $\lambda > 0$, we get a tighter sensitivity score. We exemplify this further. For simplicity, assume the number of clients to be 1, which can be easily extended to a setup with multiple clients. Let $\mathbf{X}$ be a dataset with $n$ points in $\mathbb{R}^d$ such that $n/d = c$ where $c$ is a positive integer. Again, for simplicity, in the case of ridge regression, the response vector $\mathbf{y}$ is a zero vector, and for regularized regression, it is an all 1 vector in $n$-dimensional space. Let $\mathbf{X} = \begin{bmatrix} \mathbf{I}, \cdots, \mathbf{I} \end{bmatrix} \in \mathbb{R}^{d \times n}$ where $\mathbf{I}$ is identity matrix. In (Huang et al., 2022), the sensitivity score for every point in the ridge regression problem is at least $1/c$. Hence, the total sensitivity for $n$ points is $n/c = d$, which directly affects the final coreset size. Notice that it is irrespective of the fact whether $\lambda$ is 0 or a positive scalar. So, in such a case, our sensitivity scores are $1/(c + \lambda)$. Hence, the total sensitivity score is $n/(c + \lambda) < n/c = d$. In fact, for higher values of $\lambda$, the total sensitivity score could be significantly smaller. So, theoretically, the improvement in the coreset size is at least by a factor of $c/(c + \lambda)$. For our algorithm, obtaining a tighter upper bound on these functions is sufficient.

Next, for both problems, we define a function for every point and every client such that the aggregation of the functions for every point from different clients ensures a tight upper-bound on the sensitivity of the complete high-dimensional points. These are then further used to sample points and assign appropriate weights to them. We use the unified coreset construction algorithm from (Huang et al., 2022), which we state as algorithm 3 in the appendix for completeness. Here, we describe the overview of the algorithm.

**Algorithm Overview:** The algorithm uses scores $\mathbf{g}^j = \{g_1^{(j)}, \ldots, g_n^{(j)}\}$ for every client $j \in [T]$. Next, every $j \in [T]$ shares its local sum of scores, $G^{(j)}$, with the server. Using these values, the server computes a distribution over $[T]$ and samples a set of clients $C \subseteq T$. Here, clients with higher $G^{(j)}$ will have a greater likelihood of being selected by the server. Next, it asks every selected client to sample $\lceil m/t \rceil$ points and send their indices to the server. This ensures that the union of the sampled indices forms a set $S$, with expected size of $\mathbb{E}[|S|] = m$. Finally, based on the received indices, the server determines the weight function $w$ for all the sampled indices $S$ and returns the weighted set $\mathbf{S}_w$, where $\mathbf{S} \subseteq \mathbf{Z}$ and $w : \mathbf{S} \to \mathbb{R}_{>0}$.

**Vertical Federated Optimization:** At every round $r \in [R]$, the server makes a call of Algorithm 3 to compute a coreset $\mathbf{S}_w$ of size $m$. It then informs $\mathbf{S}_w$ to the participating clients $j \in [T]$ to compute the local gradients $\nabla_{\mathbf{q}^{(j)}} \text{cost}(\mathbf{X}^{(j)}, \mathbf{q}_r^{(j)})$ using $\mathbf{S}_w$. Similar to the equation 6, the server then collects the gradients to update the model as $\mathbf{q}_{r+1} = \mathbf{q}_r - \eta \bigcup_{j \in [T]} \nabla_{\mathbf{q}^{(j)}} \text{loss}(\mathbf{S}_w^{(j)}, \mathbf{q}_r^{(j)}, \lambda)$.

## 5. Coreset Construction for VRLog

Here, we present how to compute the scores $g_i^{(j)}$ for every client $j \in [T]$ and every point $i \in [n]$ for VRLog. For simplicity, we start with the case where $T = 1$ and bound the sensitivity scores. To get a practical bound, we use a data dependent property known as $\mu$-complexity of the dataset, as introduced by (Munteanu & Schwiegelshohn, 2018).

**Definition 5.1** ($\mu$-**Complexity**). For a given dataset $\mathbf{Z} \in \mathbb{R}^{n \times d}$ and a vector $\mathbf{q} \in \mathbb{R}^d$, let $(\mathbf{Zq})^+$ and $(\mathbf{Zq})^-$ be vectors having only positive and negative entries respectively. Similarly $(\mathbf{q})^+$ and $(\mathbf{q})^-$ are defined. Then the $\mu$-complexity for the regularized logistic regression with a regularization parameter $\lambda$ is defined as,

$$\mu(\mathbf{Z}, \lambda) := \sup_{\mathbf{q} \in \mathbb{R}^d \setminus \{0\}} \frac{\|(\mathbf{Zq})^+\|_1 + \lambda \|(\mathbf{q})^+\|_1}{\|(\mathbf{Zq})^-\|_1 + \lambda \|(\mathbf{q})^-\|_1}.$$

Notice that due to $\sup_{\mathbf{q}}()$, we have $\mu(\mathbf{Z}, \lambda) \geq 1$. It implies $\mu^{-1} (\|(\mathbf{Zq})^-\|_1 + \lambda \|(\mathbf{q})^-\|_1) \leq \|(\mathbf{Zq})^+\|_1 + \lambda \|(\mathbf{q})^+\|_1 \leq \mu (\|(\mathbf{Zq})^-\|_1 + \lambda \|(\mathbf{q})^-\|_1)$. For brevity, we refer to $\mu(\mathbf{Z}, \lambda)$ simply as $\mu$ in the future.W We consider

an augmented matrix $\hat{\mathbf{Z}}^\top := (\mathbf{Z}^\top, \lambda\mathbf{I})$. In the following lemma, we show that the $\ell_2$ norm of the orthonormal column basis of $\hat{\mathbf{Z}}$ is the upper bound of the sensitivity scores.

**Lemma 5.2.** *Let $\mathbf{Z} \in \mathbb{R}^{n \times d}$ be a $\mu$-complex dataset. Let $\lambda > 0$, and $\mathbf{U}$ be an orthonormal column basis of $\hat{\mathbf{Z}}$. Then the sensitivity scores for every $i \in [n]$,*

$$\sup_{\mathbf{q} \in \mathbb{R}^d} \frac{f(\mathbf{z}_i, \mathbf{q})}{\mathsf{ClassLoss}(\mathbf{Z}, \mathbf{q}, \lambda)} \le 20(1 + \mu)\left(\sqrt{\mathbf{u}_i^\top \mathbf{u}_i} + \frac{1}{n}\right).$$

To prove the above lemma, we analyze two possible cases $\mathbf{z}_i^\top \mathbf{q} \ge 0.5$ and $\mathbf{z}_i^\top \mathbf{q} < 0.5$ for every $\mathbf{q} \in \mathbb{R}^d$. Detailed proof has been discussed in the appendix. The sensitivity scores remain tight as they effectively capture the impact of the regularization parameter.

Let, $\mathbf{A} \in \mathbb{R}^{n \times d}$ be a matrix with singular values are $\{\sigma_i\}_{i=1}^d$. Given a scalar $\lambda > 0$ the statistical dimension is defined as $sd(\mathbf{A}, \lambda, 2) = \sum_{i=1}^d \frac{1}{1 + \frac{\lambda}{\sigma_i^2}}$. Additionally, it is known that the VC dimension of logistic regression is $d + 1$. So, there is a $\varepsilon$-net of queries of size $O\left(\frac{2}{\varepsilon}\right)^d$ (Matoušek, 1993).

Using a standard coreset construction framework, for an $\varepsilon \in (0, 1)$, if the final coreset size is $O\left(\frac{\sqrt{n \cdot sd(\mathbf{Z}, \lambda, 2)} d \log(1/\varepsilon)}{\varepsilon^2}\right)$ then using Bernstein's inequality and taking a union bound over the $\varepsilon$-net, we get an $\varepsilon$-coreset for regularized logistic regression with probability 0.99.

Notice that the coreset size is still a function of $\sqrt{n}$. We get rid of this dependence due to an improved analysis of our sensitivity-based coreset construction algorithm using Lewis weights.

**Theorem 5.3.** *(Lewis, 1978) Let $\mathbf{Z}$ be $d$-dimensional column space in $\mathbb{R}^n$ and a fixed $1 \le p < \infty$. Then, there exists a basis matrix $\mathbf{U}$ that spans the column space of $\mathbf{Z}$. The matrix $\mathbf{U}$ is called $\ell_p$ Lewis Basis of $\mathbf{Z}$ if $\mathbf{D}^{p/2-1}\mathbf{U}$ is an orthonormal matrix, where $\mathbf{D}$ is a diagonal matrix such that $D_{ii} = \sqrt{\mathbf{u}_i^\top \mathbf{u}_i}$, for every $i \in [n]$.*

As Lewis basis $\mathbf{U}$ is basis for the column space spanned by $\mathbf{Z}$, hence due to row operation on $\mathbf{U}$ by a positive definite matrix $\mathbf{D}$ does not alter its column space. As a result, the orthogonal matrix $\mathbf{D}^{p/2-1}\mathbf{U}$ spans the same column space of $\mathbf{Z}$, making it an orthonormal column basis of the matrix.

In (Mai et al., 2021), it was shown that for models such as logistic regression and hinge loss, sampling points proportional to the $\ell_1$ Lewis Weights ensures coreset guarantees. For every row $i \in [n]$ its lewis weight is $\|\mathbf{u}_i\|_2^p$, where $\mathbf{u}_i$ is the $i^{th}$ row vector of $\mathbf{U}$. For a $\mu$-complex dataset $\mathbf{Z}$ the desired coreset size for unregularized logistic regression is $\tilde{O}\left(\frac{d \cdot \mu^2}{\varepsilon^2}\right)$ [Corollary 9 (Mai et al., 2021)]. The unregularized logistic regression implies that $\lambda = 0$. This is due to the fact that the logistic regression classification loss function

looks like a hinge, which is why the $\ell_1$ Lewis weights are used to preserve the sum of absolutes, i.e., the essential part of the hinge function. Since $\mathbf{Z} \in \mathbb{R}^{n \times d}$ considered to be full rank, so for $p = 1$, $\mathbf{D}^{-1/2}\mathbf{U}$ being orthonormal column basis of $\mathbf{Z}$, it ensures that the sum of $\ell_1$ Lewis weights is $\|\mathbf{U}\|_2 := \|\mathbf{D}^{-1/2}\mathbf{U}\|_2^2 = d$. This is due to the existence of the Lewis Basis (Musco et al., 2022). We give a simple proof in the appendix for completeness.

Now, for the regularized logistic regression, we focus on the augmented matrix $\hat{\mathbf{Z}} := \begin{pmatrix} \mathbf{Z} \\ \lambda\mathbf{I} \end{pmatrix}$ which is a $(n + d) \times d$ size matrix. The algorithm computes the $\ell_1$ Lewis weights for $\hat{\mathbf{Z}}$. Now, in our VRLog problem, we use the Lewis weights for computing the $g_i^{(j)}$ for every client $j \in [T]$ and every point $i \in [n]$. score for every point $i \in [n]$.

---

**Algorithm 1** Weights for VRLog

---

**Input**: Each client $j \in [T]$ holds data $\{\mathbf{X}^{(j)}, \mathbf{y}\}$ and a real number $\lambda > 0$
**Output**: Scores $\mathbf{g}^{(j)} \in \mathbb{R}_{>0}^n$
  1: Compute $\mathbf{Z}^{(j)} \in \mathbb{R}^{n \times d_j}$, from $\{\mathbf{X}^{(j)}, \mathbf{y}\}$.
  2: Compute $\hat{\mathbf{Z}}^{(j)} := \begin{pmatrix} \mathbf{Z}^{(j)} \\ \lambda\mathbf{I}_{d_j} \end{pmatrix}$
  3: **return** $\mathbf{g}^{(j)} := \mathsf{LewisWeight}(\hat{\mathbf{Z}}^{(j)}, 1)$

---

**Algorithm Overview:** The algorithm 1 considers that the augmented dataset $\hat{\mathbf{Z}}$ is feature-wise partitioned among scattered clients as governed by the original partition of $\mathbf{Z}$, where $\mathbf{z}_i^{(j)} = -y_i \cdot \mathbf{x}_i^{(j)}$ for every $i \in [n]$. It then computes the $\ell_1$ Lewis scores for all the $n + d$ rows, locally at each client. However, the algorithm only returns the first $n$ scores, which are subsequently used by the algorithm 3 to define a distribution and sample an appropriate number of points. The final coreset size depends on the $\mu$-complexity and the statistical dimension of $\mathbf{Z}$ with regularization $\lambda$ for $\ell_1$. The statistical dimension for $\ell_2$ has been defined in (Avron et al., 2017; Ranjan & Shit, 2024). For $p = 1$, we define the statistical dimension for $\ell_1$ as follows.

**Definition 5.4.** Given a matrix $\mathbf{A} \in \mathbb{R}^{n \times d}$ and a real positive value $\lambda$, let $\mathbf{U}$ be the $\ell_1$ Lewis basis of $\mathbf{A}$ and $\{\sigma_i\}_{i=1}^d$ be the singular values of $\mathbf{M}$ where $\mathbf{M} = \mathbf{U}^\top \mathbf{D}^{p/2-1}\mathbf{A}$. Then the statistical dimension for $\ell_1$ is $sd(\mathbf{A}, \lambda, 1) := \sum_{i=1}^d \frac{1}{1 + \frac{\lambda}{\sigma_i^2}}$.

The statistical dimension of a matrix in $\ell_p$ is an important parameter. One of the main results in the paper is Lemma 5.5, which is the foundation for bonding the coreset size for regularized logistic regression. We bound the total Lewis scores of the first $n$ points of the matrix $\hat{\mathbf{Z}}$.

**Lemma 5.5.** *In lemma 5.2, let $\mathbf{U}$ and $\hat{\mathbf{U}}$ be the $\ell_1$ Lewis basis of $\mathbf{Z}$ and $\hat{\mathbf{Z}}$ respectively. Then the sum of $\ell_1$ Lewis*

weights of first $n$ points in $\hat{\mathbf{U}}$ is $sd(\mathbf{Z}, \lambda, 1)$. Here $\mathbf{M} = \mathbf{U}^{\top}\mathbf{D}^{p/2-1}\mathbf{Z}$ such that $\mathbf{D}$ is the diagonal matrix defined from $\mathbf{U}$.

We prove the above lemma in the appendix. Now, using the lemma 5.5 and Corollary 9 in (Mai et al., 2021), we have the following theorem, which is our main result for regularized logistic regression in the centralized setting.

**Theorem 5.6.** *For a given* $\mathbf{Z}$*, let* $\lambda > 0$ *be a regularization parameter. Let* $\hat{\mathbf{Z}}$ *be the augmented matrix. If* $\hat{\mathbf{Z}}$ *be a* $\mu$*-complex dataset. Let algorithm 1 computes the scores* $\mathbf{g}^{(j)}$ *for every* $j \in [T]$*. Then if then the size of the returned set from algorithm 3 is* $O\left(\frac{sd(\mathbf{Z}, \lambda, 1) \cdot \mu^2}{\varepsilon^2}\right)$*, then the set if an* $\varepsilon$*-coreset for VRLog with one client.*

Lewis weights can be approximated by an iterative algorithm (Cohen & Peng, 2015). We restate it as algorithm 4 for completeness. Notice that even though it is an iterative algorithm, the weights are always non-negative. Hence, we finally get a vector $\mathbf{g}$ representing the Lewis weights of all the rows of the input matrix.

It is known from 5.2 or (Mai et al., 2021) that the sensitivity scores for logistic regression are upper bound by a function that is proportional to the $\ell_2$ norm of its corresponding in its orthonormal column basis. These are effectively the square root of the leverage scores, which is upper bounded by a function proportional to $\sqrt{n}$ (see Lemma 5.2). However, in the case when an orthonormal column basis is constructed from the Lewis basis, we get much tighter upper bounds.

Now, consider the VFL setting with $T$ clients, such that every $j \in [T]$ has access to $\mathbf{X}^{(j)} \in \mathbb{R}^{n \times d_j}$ and $\sum_{j \in [T]} d_j = d$. The sensitivity scores on the complete feature space can be upper bounded by the sum of local upper bounds and a factor that is proportional to $T$. We have discussed this in detail in the appendix.

Notice that algorithm 4 gets $p = 1$. For every client $j \in [T]$, the algorithm takes $O(nd_j^2)$ to return the Lewis weights $g_i^{(j)}$ for every $i \in [n]$. The following corollary states our coreset guarantee for VRLog.

**Corollary 5.7** (**Coresets for VRLog**)**.** *For a given dataset* $\mathbf{Z}$ *and a scalar* $\lambda > 0$*, let* $\hat{\mathbf{Z}}$ *be the augmented matrix such that it is partitioned among* $T$ *clients. For every* $j \in [T]$*, as* $\hat{\mathbf{Z}}^{(j)} \in \mathbb{R}^{n \times d_j}$*. If* $\hat{\mathbf{Z}}$ *be a* $\mu$*-complex dataset then the algorithm 3 computes an* $\varepsilon$*-coreset (see; Definition 1.3) in* $\tilde{O}(nd^2)$ *of size* $m = O\left(\frac{\mu^2 T \sum_{j=1}^{T} sd(\mathbf{Z}^{(j)}, \lambda, 1)}{\varepsilon^2}\right)$ *for some* $\varepsilon \in (0, 1)$ *and the model can be trained with communication complexity* $O(mT)$*.*

## 6. Coreset Construction for VRLR

In this section, we present an improved $\varepsilon$-coreset for VRLR compared to (Huang et al., 2022). Recall the input dataset $\mathbf{Z}$ and its partition among clients for this problem. Our algorithm uses importance sampling and follows the unified framework. We propose a new algorithm that computes a tighter bound of the novel sensitivity scores (see Definition 4.1). Through improved analysis, we not only reduce the coreset size but also eliminate the dependence on a dataset property that is influenced by the partitioning among the clients (Huang et al., 2022). This parameter can grow as large as $\left(\frac{\sigma_{\max}(\mathbf{Z})}{\sigma_{\min}(\mathbf{Z})}\right)^2$, where $\sigma_{\max}$ and $\sigma_{\min}$ are the largest and the smallest singular values of the dataset $\mathbf{Z}$. An adversary can generate a dataset where this property is arbitrarily large. In contrast, our approach ensures that the coreset size is independent of this parameter, and instead it only depends on $T$ (number of clients). Now, we present our algorithm for computing the $g_i^{(j)}$ scores for every $i \in [n]$ at each client $j \in [T]$ for the VRLR problem.

---

**Algorithm 2** Scores for VRLR

**Input**: Each client $j \in [T]$ holds data $[\mathbf{X}^{(j)}, \mathbf{y}]$ and a real number $\lambda > 0$
**Output**: Scores $\mathbf{g}^{(j)} \in \mathbb{R}^n_{>0}$
1: **if** $j == T$ **then**
2:     Compute $\hat{\mathbf{Z}}^{(T)} := \begin{pmatrix} \mathbf{X}^{(T)} & \mathbf{y} \\ \sqrt{\lambda}\mathbf{I}_{d_T} & \mathbf{0} \end{pmatrix}$
3: **else**
4:     Compute $\hat{\mathbf{Z}}^{(j)} := \begin{pmatrix} \mathbf{X}^{(j)} \\ \sqrt{\lambda}\mathbf{I}_{d_j} \end{pmatrix}$
5: **end if**
6: **return** $\mathbf{g}^{(j)} := \mathsf{LewisWeight}(\hat{\mathbf{Z}}^{(j)}, 2)$

---

**Algorithm Overview:** In algorithm 2, each client $j$ considers a partition of $\hat{\mathbf{Z}}$ as earlier. It computes a tight upper bound of the sensitivity scores for their local data $\hat{\mathbf{Z}}^{(j)}$. It computes the local leverage scores ($\ell_2$ Lewis weights) at each client $j$ for every rows of $\hat{\mathbf{Z}}^{(\mathbf{j})}$. For a tall and thin matrix, these are the squares of the $\ell_2$ norms of the rows of its orthonormal column basis. So, every client $j \in [T]$, computes the orthonormal column $\mathbf{U}^{(j)}$ for $\hat{\mathbf{Z}}^{(j)}$). These computation happens in algorithm 4 where we set $p = 2$. Then for every $i \in [n]$ and every $j \in [T]$ the algorithm $\mathsf{LewisWeight}(\cdot)$ computes a score $g_i^{(j)} = \|\mathbf{u}_i^{(j)}\|^2$. Finally it returns an $n$-dimensional vector $\mathbf{g}^{(j)}$ for every client $j \in [T]$

These scores are finally used by the algorithm 3 to sample points. The points returned by the algorithm 3 ensure the following guarantees.

**Theorem 6.1** (**Coresets for VRLR**)**.** *Let* $\mathbf{Z}$ *be the given dataset, partitioned between* $T \geq 1$ *clients and* $\varepsilon \in (0, 1)$*. The algorithm 3 returns a* $\varepsilon$*-coreset for VRLR (see; Def-*

inition *1.4) of size* $m = O\left(\frac{T \sum_{j=i}^{T} sd(\mathbf{Z}^{(j)}, \lambda, 2) \log(d)}{\varepsilon^2}\right)$ *in*

*input sparsity time* $O(nnz(\hat{\mathbf{Z}}))$ *such that with probability at least* 0.99. *A model can be trained on this coreset with a communication complexity* $O(mT)$.

We prove the above theorem using multiple lemmas. Our first lemma is one of the important lemmas that gives a tight upper bound on the sensitivity scores in a VFL setup.

**Lemma 6.2.** *For every point* $i \in [n]$ *and client* $j \in [T]$, *the scores returned by the Algorithm 4 for VRLR,* $g_i^{(j)} = \|\mathbf{u}_i^{(j)}\|_2^2$. *Let* $\mathbf{U}^{(1)}, \mathbf{U}^{(2)}, \ldots, \mathbf{U}^{(T)}$ *be the orthonormal column basis of* $\hat{\mathbf{Z}}^{(1)}, \hat{\mathbf{Z}}^{(2)}, \ldots, \hat{\mathbf{Z}}^{(T)}$ *respectively, then every point* $i \in [n]$ *the regularized sensitivity scores can be upper bonded as,*

$$\sup_{\mathbf{q}} \frac{(\mathbf{x}_i^\top \mathbf{q} - y_i)^2}{\|\mathbf{X}\mathbf{q} - \mathbf{y}\|_2^2 + \lambda\|\mathbf{q}\|_2^2} \leq T \cdot \left(\sum_{j=1}^{T} \|\mathbf{u}_i^{(j)}\|_2^2\right)$$

It is important to note that the bound on the sensitivity scores of the points for ridge regression is a function of $T$, i.e., the total number of clients and the aggregation of locally computed leverage scores. To compute these scores, the algorithm computes a thin SVD of $\hat{\mathbf{Z}}^{(j)}$. This is the most computationally expensive operation, which takes $O(nd_j^2)$ time. However, this running time can be significantly improved using randomization techniques. Leverage scores can be approximately computed in input sparsity time, i.e., $O(nnz(\hat{\mathbf{Z}}^{(j)}))$ (Woodruff et al., 2014).

Our next result gives a tight bound on the total sensitivity scores. Like VRLog, the total sensitivity is a function of the regularization parameter $\lambda$ and $\hat{\mathbf{Z}}$.

**Lemma 6.3.** *For the given regularization parameter* $\lambda$, *the total sensitivity scores or the sum of the sensitivity scores in the VFL setup with* $[T]$ *clients are upper bounded by* $O\left(T \cdot \sum_{j=1}^{T} sd(\mathbf{Z}^{(j)}, \lambda, 2)\right)$.

For every $j \in [T]$ the $sd(\mathbf{Z}^{(j)}, \lambda, 2)$ is the $\ell_2$ statistical dimension of $\mathbf{Z}^{(j)}$ with respect to $\lambda$. As the $\lambda$ increases, the statistical dimension decreases, thereby decreasing the bound on the total sensitivity score. The proof is discussed in the appendix. Finally, with the following lemma, we prove that the approximation guarantee holds along every direction in the complete feature space.

**Lemma 6.4.** *For a given* $\mathbf{Z} \in \mathbb{R}^{n \times (d+1)}$ *be the augmented matrix, let* $\lambda > 0$ *be a scalar and* $\varepsilon \in (0, 1)$. *The algorithm 3 samples a set* $\mathbf{S} \subseteq \mathbf{Z}$ *with appropriate weights* $w : \mathbf{S} \to \mathbb{R}_{>0}$. *We represent the weighted set as* $\mathbf{S}_w$. *If the size* $\mathbf{S}$ *is at least* $O\left(\frac{T \sum_{j=1}^{T} sd(\mathbf{Z}^{(j)}, \lambda, 2) \log(d)}{\varepsilon^2}\right)$ *then the set ensures the following guarantee with at least* 0.99 *probability.*

$$(1 - \varepsilon)(\mathbf{Z}^\top \mathbf{Z} + \lambda\mathbf{I}) \preceq \mathbf{S}_w^\top \mathbf{S}_w + \lambda\mathbf{I} \preceq (1 + \varepsilon)(\mathbf{Z}^\top \mathbf{Z} + \lambda\mathbf{I})$$

Here, we use Matrix Bernstein's inequality (Tropp et al., 2015; Chhaya et al., 2020a) to prove the above lemma, which has been deferred to the appendix. The above lemma proves that difference between the covariances of the coreset and the full dataset along with the regularization parameter is PSD bounded, i.e, $(1+\varepsilon)(\mathbf{Z}^\top \mathbf{Z} + \lambda\mathbf{I}) - \mathbf{S}_w^\top \mathbf{S}_w + \lambda\mathbf{I} \succeq 0$ and $\mathbf{S}_w^\top \mathbf{S}_w + \lambda\mathbf{I} - (1 - \varepsilon)(\mathbf{Z}^\top \mathbf{Z} + \lambda\mathbf{I}) \succeq 0$. As, it ensures ridge $\ell_2$ subspace embedding, hence for every query vector $\mathbf{q} \in \mathbb{R}^d$ we have, $\left|\|\mathbf{Z}\mathbf{q}\|_2^2 - \|\mathbf{S}_w\mathbf{q}\|_2^2\right| \leq \varepsilon\left(\|\mathbf{Z}\mathbf{q}\|_2^2 + \lambda\|\mathbf{q}\|_2^2\right)$. Finally, it ensures the desired guarantee in Theorem 6.1.

# 7. Experiments

We have conducted experiments for both regularized logistic and regularized linear regression [1]. We have considered three datasets: (1) Credit Card for VRLog problem, (2) Financial, and (3) Blog Feedback for VRLR. We have first partitioned each dataset into a training and a testing set (80:20). Further, for both problems, we have considered the number of clients to be 3, i.e., $T = 3$. We compare the performance of our coresets with various other sampling techniques. Once we have a sample from one of the sampling methods (including ours), we train an appropriate model (i.e., either regularized logistic or ridge regression). Next, we use this model to compute the training loss, test accuracy, model closeness, and training time for the VRLog experiment. For VRLR, we have reported test RMSE and model closeness. We have repeated each experiment 10 times for every sample size and reported their medians.

**VRLog:** We have considered Credit Card data, which is a binary class dataset, with imbalanced class sizes. Our algorithm is AugLewis (Algorithm 1), and the rest of the sampling methods are– (1) **Uniform:** points sampled uniformly at random. (2) **HLSZ:** Points are sampled based on the sampling method in (Huang et al., 2022). (3) **SqLev:** it is a heuristic sampling method, where the dataset is partitioned into two sets based on the labels. From both sets, we use the square root of leverage scores for sampling points (Munteanu et al., 2018). (4) **Lewis:** Uses $\ell_1$ Lewis weights of only dataset for sampling points.

In Figure 1, we have reported the training loss, balanced accuracy on test data, model closeness, and training time. Recall that in Corollary 5.7 our coreset ensures a $\varepsilon$ approximation guarantee on the training loss. The leftmost plot corroborates our theoretical guarantees.

We further observe that our coreset gives balanced accuracy on par with the test dataset. Even though there are no known theoretical guarantees for models trained on coresets and models trained on the complete data, we observe that

---

[1]Codes available at https://github.com/dcll-iiitd/CoresetForVFL

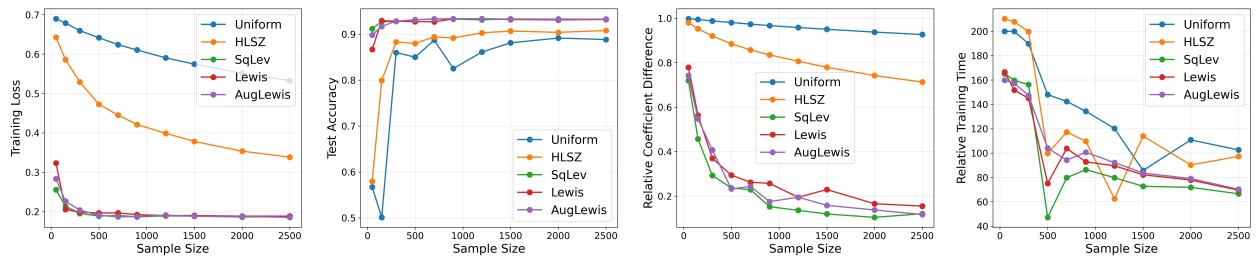

*Figure 1.* VRLog Coreset Performance (Credit Card)

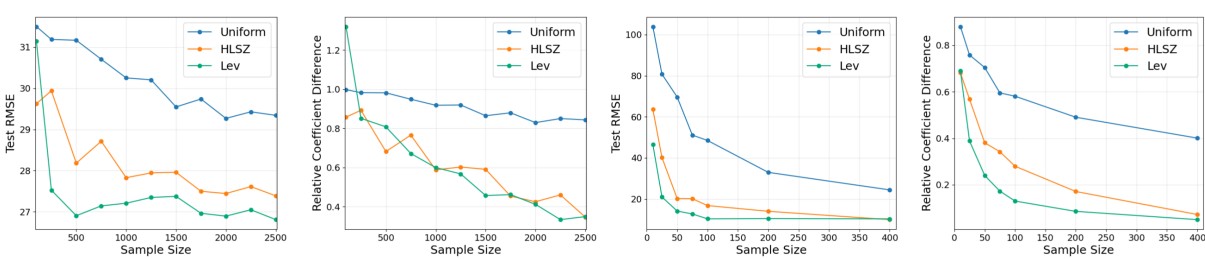

*Figure 2.* VRLR Coreset Performance (Blog Feedback, Financial Dataset)

the model trained on our coresets is closer to the model trained on the complete data, compared to a model trained on the HLSZ sample, only SqLev outperforms. Overall, our coresets are either matching or outperforming in terms of balanced accuracy on the test dataset compared to other coreset construction methods. The SqLev and Lewis algorithms are very close to the AugLewis. However, these algorithms have poorer theoretical guarantees, such as the sizes of the coreset are $O(\sqrt{n})$ and $O(\mu^2 d^2 \varepsilon^{-2})$, respectively. In terms of training time, with our coresets training a model is around 80x to 100x faster compared to training a model on the complete dataset.

| Samples→ | 500 | | 2500 | |
|---|---|---|---|---|
| **Methods↓** | Train | Test | Train | Test |
| Uniform | 0.8192 | 0.8185 | 0.8723 | 0.8731 |
| HLSZ | 0.8704 | 0.8712 | 0.9071 | 0.9078 |
| Lewis | 0.9220 | 0.9230 | 0.9304 | 0.9315 |
| AugLewis | **0.9330** | **0.9343** | **0.9319** | **0.9331** |

*Table 1.* F1 scores on the Credit Card dataset.

We also compared the F1 scores between all the sampling methods on the Credit card datasets. We observe in table 1 that even though there are no known theoretical claims, from our or any other coreset for logistic regression, our algorithm is always better than other sampling methods for various sample sizes.

Based on both empirical evidence and established theoretical guarantees, our algorithm 1, which leverages regularized sensitivity scores, offers greater reliability and superiority

in constructing coresets for VRLog problems.

**VRLR:** For the VRLR problem, we used the Blog Feedback dataset and financial data. We compare our sampling method (Algorithm 2) with a naive Uniform sampling and Leverage Score sampling (Huang et al., 2022). In Figure 2, we reported the test RMSE and the model closeness between models trained on the coreset and a model trained on the complete dataset. In both parameters, our algorithm (**Lev**) clearly outperforms the other sampling methods, which are uniform sampling and the sampling method from (Huang et al., 2018). It verifies our theoretical claim that using regularized sensitivity scores (see Definition 4.1), our sampling method achieves smaller RMSE and parameter closeness compared to others. Hence, Lev is superior to its competitors for the VRLR problem.

## Conclusion

In this paper, we highlight the advantages of using coresets in Vertical Federated Learning (VFL). We introduce smaller coresets for regularized logistic regression in both centralized and VFL settings. Additionally, we demonstrate how a global guarantee on loss functions, utilizing the full feature space, can be achieved while maintaining data privacy among clients. We further enhance the coreset size for regularized linear regression in VFL, making it independent of data-dependent parameters. Notably, as the regularization parameter $\lambda$ increases, model complexity decreases—a trend observed in both coresets. This relationship was empirically validated through our experiments.

## Acknowledgment

Supratim acknowledges the kind support from Data-Heroes Ltd and Anusandhan National Research Foundation (ECRG/2024/000959). The work was partly supported by their generous fund.

Bapi acknowledges support in part by the Indo-French Centre for the Promotion of Advanced Research (IFC-PAR/CEFIPRA) through the FedAutoMoDL project and the Infosys Center for Artificial Intelligence (CAI) at IIIT-Delhi through the Scalable Federated Learning project. He also acknowledges support by Anusandhan National Research Foundation under project SRG/2022/002269.

## Impact Statement

The paper presents an advancement in coreset construction algorithms problems vertical federated learning. There are no societal impacts that need any special mention. ¿¿¿¿¿¿ be9d18e5cfa80e3f7e3cf3687ae0934dba330108

## Acknowledgment

## Impact Statement

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

# Appendix

Here we present all the missing (or known) algorithms for completeness and discuss all the proofs that are missing in the main paper.

## A. Missing Algorithms

Here we state the known algorithms used in our coreset construction for completeness.

### A.1. Unified Framework

The unified framework from (Huang et al., 2022) is as follows.

---
**Algorithm 3** Unified Coreset for VFL

---
**Input**: Each client $j \in [T]$ has data $\mathbf{Z}^{(j)}$, a vector $\mathbf{g}^{(j)} \in \mathbb{R}^n$, an integer $m \geq 1$ for coreset size.
**Output**: Weighted Set $\mathbf{S}_w$

1: Each client $j \in [T]$ sends $G^{(j)} := \sum_{i \in [n]} g_i^{(j)}$ to the server.
2: The server computes $G := \sum_{j \in [T]} G^{(j)}$ and samples a client subset $C \subseteq [T]$ of size $t$, where each client $j \in [T]$ is sampled with a probability $\frac{G^{(j)}}{G}$.
3: Each client $j \in C$, samples a subset $S^{(j)} \subseteq [n]$ of size $\lceil m/t \rceil$, where each point $i \in [n]$ is sampled with a probability $\frac{g_i^{(j)}}{G^{(j)}}$, and sends $S^{(j)}$ to the server.
4: The server broadcasts $S \leftarrow \bigcup_{j \in [T]} S^{(j)}$ to all parties.
5: Each client $j \in C$ sends $\left\{ g_i^{(j)} : i \in S \right\}$ to the server.
6: For every point $i \in S$, server computes weights $w(i) \leftarrow \frac{G}{|S| \cdot \sum_{j \in [T]} g_i^{(j)}}$.
7: **return** weighted set $\mathbf{S}_w$

---

### A.2. Lewis Weights

The algorithm to compute Lewis Weights (Cohen & Peng, 2015) is as follows.

---
**Algorithm 4** LewisWeight

---
**Input**: A matrix $\mathbf{X}$, an integer $p \in \{1, 2\}$
**Output**: $\mathbf{g} \in \mathbb{R}^n$

1: $n = \#row(\mathbf{X})$
2: $\mathbf{W} = \mathbf{I}_n$
3: **for** $t = 1 \dots 10$ **do**
4:     **for** $i = 1 \dots n$ **do**
5:         Set $W_{ii} \leftarrow \left( \mathbf{x}_i^\top (\mathbf{X}^\top \mathbf{W}^{1-2/p} \mathbf{X})^{-1} \mathbf{x}_i \right)^{\frac{p}{2}}$.
6:     **end for**
7: **end for**
8: **return** $\mathbf{g} := \text{diag}(\mathbf{W})$

---

## B. Proofs of VRLog

### B.1. Proof of Lemma 5.2

**Lemma B.1.** *Let $\mathbf{Z} \in \mathbb{R}^{n \times d}$ be a $\mu$-complex dataset. Let $\lambda > 0$, and $\mathbf{U}$ be an orthonormal column basis of $\hat{\mathbf{Z}}$. Then the sensitivity scores for every $i \in [n]$,*

$$\sup_{\mathbf{q} \in \mathbb{R}^d} \frac{f(\mathbf{z}_i, \mathbf{q})}{\mathsf{ClassLoss}(\mathbf{Z}, \mathbf{q}, \lambda)} \leq 200(1 + \mu) \left( \sqrt{\mathbf{u}_i^\top \mathbf{u}_i} + \frac{1}{n} \right).$$

*Proof.* Let $i \in [n]$ and $\mathbf{q}_i \in \arg\max_{\mathbf{q}} \frac{f(\mathbf{z}_i, \mathbf{q})}{\mathsf{ClassLoss}(\mathbf{Z}, \mathbf{q}, \lambda)}$. We prove the theorem by considering two cases.

1. $\mathbf{z}_i^T \mathbf{q}_i \geq 0.5$

2. $\mathbf{z}_i^T \mathbf{q}_i < 0.5$

**Case 1:** $\mathbf{z}_i^T \mathbf{q}_i \geq 0.5$

*Proof.* Consider QR decomposition of $\hat{\mathbf{Z}}$ as $\hat{\mathbf{Z}} = \mathbf{UR}$. Here $\mathbf{U}$ is an orthonormal basis for the column space of $\hat{\mathbf{Z}}$. When $0.5 \leq \mathbf{z}_i^\top \mathbf{q}$ and monotonicity of $f$ that

$$
\begin{aligned}
f(\mathbf{z}_i^\top \mathbf{q}) &= f\left(\mathbf{z}_i^\top \mathbf{q}\right) \\
&= f\left(\mathbf{u}_i^\top \mathbf{Rq}\right) \\
&\overset{(i)}{\leq} f\left(\|\mathbf{u}_i\|_2 \|\mathbf{Rq}\|_2\right) \\
&\overset{(ii)}{=} f\left(\|\mathbf{u}_i\|_2 \|\mathbf{URq}\|_2\right) \\
&= f\left(\|\mathbf{u}_i\|_2 \|\hat{\mathbf{Z}}\mathbf{q}\|_2\right) \\
&\overset{(iii)}{\leq} 2\|\mathbf{u}_i\|_2 \|\hat{\mathbf{Z}}\mathbf{q}\|_2 \\
&\leq 2\|\mathbf{u}_i\|_2 \|\hat{\mathbf{Z}}\mathbf{q}\|_1 \\
&\overset{(iv)}{\leq} 2\|\mathbf{u}_i\|_2 (1+\mu) \|(\hat{\mathbf{Z}}\mathbf{q})^+\|_1 \\
&\leq 2\|\mathbf{u}_i\|_2 (1+\mu) \left( \sum_{j:\mathbf{z}_j^\top \mathbf{q} \geq 0} f(\mathbf{z}_j^\top \mathbf{q}) + \lambda |\mathbf{q}^{(+)}| \right) \\
&\leq 2\|\mathbf{u}_i\|_2 (1+\mu) \mathsf{ClassLoss}(\mathbf{Z}, \mathbf{q}, \lambda).
\end{aligned}
\tag{7}
$$

The inequality $(i)$ is due to Cauchy Schwarz. Since $\mathbf{U}$ is an orthonormal matrix which is invariant towards $\ell_2$ so we have the equality $(ii)$. For a sufficiently large $|a|$, we have $|a| \leq f(a) \leq 2|a|$; due to this, we get the inequality $(iii)$. The inequality is from the $\mu$-complexity of $\hat{\mathbf{Z}}$. Finally we get the equation (7). $\qquad \square$

**Case 1:** $\mathbf{z}_i^T \mathbf{q}_i < 0.5$

*Proof.* Let $K^- = \{j \in [n] \mid \mathbf{z}_j^\top \mathbf{q}) \leq -2\}$ and $K^+ = \{j \in [n] \mid \mathbf{z}_j^\top \mathbf{q} > -2\}$. Note that $f(-2) > 1/100$ and $f(\mathbf{z}_j^\top \mathbf{q}) \leq f(0.5) < 1$. Also, $n = |K^+| + |K^-|$. Thus if $|K^+| \geq \frac{n}{2}$ then

$$
\begin{aligned}
\mathsf{ClassLoss}(\mathbf{Z}, \mathbf{q}, \lambda) &= \sum_{i=1}^{n} f(\mathbf{z}_j^\top \mathbf{q}) + \lambda \|\mathbf{q}\|_1 \\
&\overset{(i)}{\geq} \frac{n}{200} \\
&\geq \frac{n}{200} \cdot f(\mathbf{x}_i^\top \mathbf{q}).
\end{aligned}
\tag{8}
$$

We have the inequality $(i)$ because $\mathbf{z}_j^\top \mathbf{q} > -2$ and such cases $f(-2) \geq 1/100$ further $|K^+| \geq n/2$. Now, if $|K^+| < \frac{n}{2}$ then $|K^-| \geq \frac{n}{2}$. Further, $f(\mathbf{z}_j^\top \mathbf{q}) \leq 1$ so we get the equation (8).

$$
\begin{aligned}
\mathsf{ClassLoss}(\mathbf{Z}, \mathbf{q}, \lambda) &\geq \|(\mathbf{Zq})^+\|_1 + \lambda \|\mathbf{q}^{(+)}\|_1 \\
&\overset{(i)}{\geq} \frac{\|(\mathbf{Zq})^-\|_1 + \lambda \|\mathbf{q}^{(-)}\|_1}{\mu} \\
&\geq n/(2\mu) \\
&\geq \frac{n}{2\mu} \cdot f(\mathbf{z}_i^\top \mathbf{q}).
\end{aligned}
\tag{9}
$$

The inequality $(i)$ is due to the $\mu$-complexity property of $\hat{\mathbf{Z}}$, and the following two inequality is due to the same reason from the previous analysis. $\qquad \square$

So, using the equations (7), (8) and (9) we have the following claim for every $\mathbf{q} \in \mathbb{R}^d$

$$\frac{f(\mathbf{z}_i, \mathbf{q})}{\mathsf{ClassLoss}(\mathbf{Z}, \mathbf{q}, \lambda)} \leq 200(1 + \mu) \left( \sqrt{\mathbf{u}_i^\top \mathbf{u}_i} + \frac{1}{n} \right)$$

$\qquad \square$

**Lemma B.2.** *Given a Lewis basis* $\mathbf{U}$ *of* $\mathbf{Z}$ *for some fixed p, we have* $\|\mathbf{D}^{p/2-1}\mathbf{U}\|_F^2 = \|\mathbf{U}\|_2^p$.

*Proof.* We know $D_{ii} = \sqrt{\mathbf{u}_i^\top \mathbf{u}_i}$ where $\mathbf{D}$ is a diagonal matrix. So,

$$\mathbf{D} = \mathrm{diag}\left( \sqrt{\mathbf{u}_1^\top \mathbf{u}_1}, \sqrt{\mathbf{u}_2^\top \mathbf{u}_2}, \ldots, \sqrt{\mathbf{u}_n^\top \mathbf{u}_n} \right) \text{ and subsequently we have, } \mathbf{D}^{\frac{p}{2}-1} = \mathrm{diag}\left( (\mathbf{u}_1^\top \mathbf{u}_1)^{\frac{\frac{p}{2}-1}{2}}, \ldots, (\mathbf{u}_n^\top \mathbf{u}_n)^{\frac{\frac{p}{2}-1}{2}} \right)$$

Now compute $\|\mathbf{D}^{\frac{p}{2}-1}\mathbf{U}\|_F^2$

$$\|\mathbf{D}^{\frac{p}{2}-1}\mathbf{U}\|_F^2 = \sum_{i=1}^n \|\mathbf{e}_i^\top \mathbf{D}^{p/2-1}\mathbf{U}\|_2^2 = \sum_{i=1}^n \left[ (\mathbf{u}_i^\top \mathbf{u}_i)^{p/2S-1}(\mathbf{u}_i^\top \mathbf{u}_i) \right] = \sum_{i=1}^n (\mathbf{u}_i^\top \mathbf{u}_i)^{\frac{p}{2}} = \|\mathbf{U}\|_2^p$$

$\qquad \square$

### B.2. Proof of Lemma 5.5

**Lemma B.3.** *In lemma 5.2, let* $\mathbf{U}$ *and* $\hat{\mathbf{U}}$ *be the* $\ell_1$ *Lewis basis of* $\mathbf{Z}$ *and* $\hat{\mathbf{Z}}$ *respectively. Then the sum of* $\ell_1$ *Lewis weights of first* $n$ *points in* $\hat{\mathbf{U}}$ *is* $sd(\mathbf{Z}, \lambda, 1)$. *Here* $\mathbf{M} = \mathbf{U}^\top \mathbf{D}^{p/2-1}\mathbf{Z}$ *such that* $\mathbf{D}$ *is the diagonal matrix defined from* $\mathbf{U}$.

*Proof.* Let $\hat{\mathbf{U}}$ be the Lewis Basis of the matrix $\hat{\mathbf{Z}}$ and $\mathbf{U}$ is the Lewis Basis of $\mathbf{Z}$. Let $\hat{\mathbf{D}}$ and $\mathbf{D}$ be the diagonal matrices defined from $\hat{\mathbf{U}}$ and $\mathbf{U}$. From the above lemma B.2 we know that $\|\hat{\mathbf{D}}^{-1/2}\hat{\mathbf{U}}\|_F^2 = \|\hat{\mathbf{U}}\|_2$. Hence, in the regularized logistic regression, the total Lewis weights is $\sum_{i=1}^n \|\hat{\mathbf{u}}_i\|_1 = \sum_{i=1}^n \hat{D}_{ii}^2 \hat{\mathbf{u}}_i^\top \hat{\mathbf{u}}_i$.

Let $\mathbf{M} = \mathbf{U}^\top \mathbf{D}^{-1/2}\mathbf{Z}$ a $d \times d$ full rank matrix, such that its decomposition is $\mathbf{M} = \tilde{\mathbf{U}}\tilde{\Sigma}\tilde{\mathbf{V}}^\top$. Now, consider a matrix $\mathbf{N} = \begin{pmatrix} \mathbf{D}^{-1/2}\mathbf{U}\tilde{\mathbf{U}}\tilde{\Sigma}\Sigma \\ \sqrt{\lambda}\tilde{\mathbf{V}}\Sigma \end{pmatrix}$ where $\Sigma = \left( \tilde{\Sigma}^2 + \lambda \mathbf{I}_d \right)^{-1/2}$. It is not difficult to verify that $\mathbf{N}^\top \mathbf{N} = \mathbf{I}_d$. Now, recall that the column space of $\hat{\mathbf{Z}}$ is same as the column space of both $\hat{\mathbf{U}}$ and $\hat{\mathbf{D}}^{-1/2}\hat{\mathbf{U}}$. Further, $\mathbf{N}$ is a column basis of $\hat{\mathbf{Z}}$ and $\mathbf{N}$ is an orthonormal matrix. Hence, $\mathbf{N}$ is an orthonormal column basis of $\hat{\mathbf{Z}}$. Since, $\mathbf{N}$ and $\hat{\mathbf{D}}^{-1/2}\hat{\mathbf{U}}$ both are the orthonormal column basis of $\hat{\mathbf{Z}}$, so they are a rotation apart from each other. As Frobenius norm is invariant to rotations hence, we have $\sum_{i=1}^n \|\hat{\mathbf{u}}_i\|_1 = \sum_{i=1}^n \hat{D}_{ii}^{-1}\hat{\mathbf{u}}_i^\top \hat{\mathbf{u}}_i = \sum_{i=1}^n \mathbf{n}_i^\top \mathbf{n}_i$ where $\mathbf{n}_i^\top$ is the $i^{th}$ row of $\mathbf{N}$.

Now, we bound $\sum_{i=1}^n \mathbf{n}_i^\top \mathbf{n}_i$.

$$\begin{aligned} \sum_{i=1}^n \mathbf{n}_i^\top \mathbf{n}_i &= \|\mathbf{D}^{-1/2}\mathbf{U}\tilde{\mathbf{U}}\tilde{\Sigma}\Sigma\|_F^2 \\ &\overset{(i)}{=} \|\tilde{\Sigma}\Sigma\|_F^2 \\ &\overset{(ii)}{=} \sum_{i=1}^d \frac{\tilde{\sigma}_i^2}{\tilde{\sigma}_i^2 + \lambda}. \end{aligned}$$

Here, $(i)$ is because both $\mathbf{D}^{-1/2}\mathbf{U}$ and $\tilde{\mathbf{U}}$ are the orthonormal column basis, to which the Frobenius norm is invariant. Since both $\Sigma$ and $\tilde{\Sigma}$ are diagonal matrices, we get the final equality $(ii)$. $\qquad \square$

### B.3. Proof of Corollary 5.7

**Corollary B.4.** *For a given dataset $\mathbf{Z}$ and a scalar $\lambda > 0$, let $\hat{\mathbf{Z}}$ be the augmented matrix such that it is partitioned among $T$ clients. For every $j \in [T]$, as $\hat{\mathbf{Z}}^{(j)} \in \mathbb{R}^{n \times d_j}$. If $\hat{\mathbf{Z}}$ be a $\mu$-complex dataset then the algorithm 3 computes an $\varepsilon$-coreset (see; Definition 1.3) in $\tilde{O}(nd^2)$ of size $m = O\left(\frac{\mu^2 T \sum_{j=1}^{T} sd(\mathbf{Z}^{(j)}, \lambda, 1)}{\varepsilon^2}\right)$ for some $\varepsilon \in (0, 1)$ and the model can be trained with communication complexity $O(mT)$.*

*Proof.* For logistic regression from Lemma 5.2 we know that the sensitivity scores for every point $i \in [n]$ can be upper bounded by a function that is proportional to the $\ell_2$ norm of the row of any orthonormal column basis of the matrix. Further, from (Mai et al., 2021) we know due to the existence of Lewis Basis, there is a tighter upper bound that is proportional to the square of the $\ell_2$ norm of a special orthonormal column basis of dataset that is constructed from its Lewis Basis.

So, for a matrix $\mathbf{A} \in \mathbb{R}^{n \times d}$, the higher upper bound for any $i \in [n]$ is $\max_{\mathbf{q} \in \mathbb{R}^d} \frac{(\mathbf{a}_i \top \mathbf{q})^2}{\|\mathbf{A}\mathbf{q}\|_2^2}$. Now, in the case of VRLog, we can upper bound these scores as, $\max_{\mathbf{q} \in \mathbb{R}^d} \frac{(\mathbf{a}_i \top \mathbf{q})^2}{\|\mathbf{A}\mathbf{q}\|_2^2} \leq d \sum_{j=1}^{d} \left(\frac{(a_{ij} \cdot q_j)^2}{\mathbf{e}_j^\top \mathbf{A}^\top \mathbf{A} \mathbf{e}_j \cdot q_j^2}\right) = d \sum_{j=1}^{d} \left(\frac{a_{ij}^2}{\mathbf{e}_j^\top \mathbf{A}^\top \mathbf{A} \mathbf{e}_j}\right)$. We get this bound by applying Cauchy Schwarz in the numerator and the in the denominator we use a lower bound. We use $\|\mathbf{A}\mathbf{q}\|_2^2 \geq \mathbf{e}_j^\top \mathbf{A}^\top \mathbf{A} \mathbf{e}_j \cdot q_j^2$ for every $j \in [d]$.

Now, recall that due to lemma B.2, the coreset size depends on the $\|\mathbf{D}^{-1/2}\mathbf{U}\|_2$. For every point $i \in [n]$, its sensitivity score is upper bounded by the $\|\mathbf{e}_i^\top \mathbf{D}^{-1/2}\mathbf{U}\|_2$. Hence, the sum of the Lewis weights from different clients and factor of $T$ upper bounds the actual Lewis weight of the point in higher dimensional space. $\square$

## C. Proofs of VRLR

### C.1. Proof of Lemma 6.2

**Lemma C.1.** *For every point $i \in [n]$ and client $j \in [T]$, the scores returned by the Algorithm 4 for VRLR, $g_i^{(j)} = \|\mathbf{u}_i^{(j)}\|_2^2$. Let $\mathbf{U}^{(1)}, \mathbf{U}^{(2)}, \ldots, \mathbf{U}^{(T)}$ be the orthonormal column basis of $\hat{\mathbf{Z}}^{(1)}, \hat{\mathbf{Z}}^{(2)}, \ldots, \hat{\mathbf{Z}}^{(T)}$ respectively, then every point $i \in [n]$ the regularized sensitivity scores can be upper bonded as,*

$$\sup_{\mathbf{q}} \frac{(\mathbf{x}_i^\top \mathbf{q} - y_i)^2}{\|\mathbf{X}\mathbf{q} - \mathbf{y}\|_2^2 + \lambda\|\mathbf{q}\|_2^2} \leq T \cdot \left(\sum_{j=1}^{T} \|\mathbf{u}_i^{(j)}\|_2^2\right)$$

*Proof.* For every point $i$, the regularized sensitivity function is defined as, $\sup_{\mathbf{q}} \frac{(\mathbf{x}_i^\top \mathbf{q} - y_i)^2}{\|\mathbf{X}\mathbf{q} - \mathbf{y}\|_2^2 + \lambda\|\mathbf{q}\|_2^2}$. For simplicity, assume that $T = d$ and $\mathbf{y} = \mathbf{0}$, such that every client $j \in [T]$, has access to $\mathbf{x}_j$, which is the $j^{th}$ column of $\mathbf{X}$. We consider $\mathbf{Z} = \mathbf{X}$ and $\hat{\mathbf{Z}} = \begin{pmatrix} \mathbf{Z} \\ \sqrt{\lambda}\mathbf{I} \end{pmatrix}$. Now we analyze the sensitivity score for every $\mathbf{q}$ without the supremum as follows.

$$
\begin{aligned}
\frac{(\mathbf{x}_i^\top \mathbf{q} - y_i)^2}{\|\mathbf{X}\mathbf{q} - \mathbf{y}\|_2^2 + \lambda\|\mathbf{q}\|_2^2} &= \frac{(\mathbf{x}_i^\top \mathbf{q})^2}{\|\mathbf{X}\mathbf{q}\|_2^2 + \lambda\|\mathbf{q}\|_2^2} \\
&= \frac{(\mathbf{z}_i^\top \mathbf{q})^2}{\|\hat{\mathbf{Z}}\mathbf{q}\|_2^2} \\
&= \frac{(z_{i1} \cdot q_1 + z_{i2} \cdot q_2 + \ldots + z_{id} \cdot q_d)^2}{\|\hat{\mathbf{Z}}\mathbf{q}\|_2^2} \\
&\stackrel{(i)}{\leq} \frac{d\left((z_{i1} \cdot q_1)^2 + (z_{i2} \cdot q_2)^2 + \ldots + (z_{id} \cdot q_d)^2\right)}{\|\hat{\mathbf{Z}}\mathbf{q}\|_2^2} \\
&\stackrel{(ii)}{\leq} d\left(\frac{(z_{i1} \cdot q_1)^2}{\mathbf{e}_1^\top \hat{\mathbf{Z}}^\top \hat{\mathbf{Z}} \mathbf{e}_1 \cdot q_1^2} + \frac{(z_{i2} \cdot q_2)^2}{\mathbf{e}_2^\top \hat{\mathbf{Z}}^\top \hat{\mathbf{Z}} \mathbf{e}_2 \cdot q_2^2} + \ldots + \frac{(z_{id} \cdot q_d)^2}{\mathbf{e}_d^\top \hat{\mathbf{Z}}^\top \hat{\mathbf{Z}} \mathbf{e}_d \cdot q_d^2}\right) \\
&\stackrel{(iii)}{=} d\left(\frac{(z_{i1})^2}{\mathbf{e}_1^\top \hat{\mathbf{Z}}^\top \hat{\mathbf{Z}} \mathbf{e}_1} + \frac{(z_{i2})^2}{\mathbf{e}_2^\top \hat{\mathbf{Z}}^\top \hat{\mathbf{Z}} \mathbf{e}_2} + \ldots + \frac{(z_{id})^2}{\mathbf{e}_d^\top \hat{\mathbf{Z}}^\top \hat{\mathbf{Z}} \mathbf{e}_d}\right)
\end{aligned}
$$

The inequality $(i)$ is due to Cauchy Schwarz in the numerator. In inequality $(ii)$, we use the lower bound in the denominator, i.e., $\|\hat{\mathbf{Z}}\mathbf{q}\|_2^2 \geq \mathbf{e}_j^\top \hat{\mathbf{Z}}^\top \hat{\mathbf{Z}}\mathbf{e}_j \cdot q_j^2$ for every $j \in [d]$. Finally, we get $(iii)$. Since it is independent of $\mathbf{q}$, so it upper bounds the above function even with a supremum over $\mathbf{q}$. Here, every client $j \in [d]$ upper bounds their own function by $\frac{(z_{ij})^2}{\mathbf{e}_j^\top \hat{\mathbf{Z}}^\top \hat{\mathbf{Z}}\mathbf{e}_j}$ and sensitivity scores entire point is upper bounded by aggregating these scores and scaling it with $d$.

Notice that if $T = 1$, then we do not need to apply Cauchy Schwarz in $(i)$ since there is only one client, and it has access to the complete data. So instead of $(iii)$ we could upper bound the above function by $\mathbf{z}_i^\top (\hat{\mathbf{Z}}^\top \hat{\mathbf{Z}})^\dagger \mathbf{z}_i$. This is also equal to the square of the $\ell_2$ norm of the $i^{th}$ row of the orthonormal column basis of $\hat{\mathbf{Z}}$. So, $\mathbf{z}_i^\top (\hat{\mathbf{Z}}^\top \hat{\mathbf{Z}})^\dagger \mathbf{z}_i = \|\mathbf{u}_i\|_2^2$ where $\mathbf{u}_i$ is the $i^{th}$ row of $\mathbf{U}$ which is the orthonormal column basis of $\hat{\mathbf{Z}}$.

In a similar manner, when $1 < T < d$ every client $j \in [T]$ upper bounds its scores by $g_i^{j)} = (\mathbf{z}_i^{(j)})^\top ((\hat{\mathbf{Z}}^{(j)})^\top (\hat{\mathbf{Z}}^{(j)}))^\dagger (\mathbf{z}_i^{(j)})$. Here $g_i^{(j)}$ are the values that were returned by LewisWeight. Finally, the sensitivity score of the entire point is upper bound by aggregating these scores and scaling them with $T$.

So, for a general $T$ we $\mathbf{y} \neq \mathbf{0}$ we have,

$$\sup_{\mathbf{q}} \frac{(\mathbf{x}_i^\top \mathbf{q} - y_i)^2}{\|\mathbf{X}\mathbf{q} - \mathbf{y}\|_2^2 + \lambda\|\mathbf{q}\|_2^2} \leq T \cdot \left( \sum_{j=1}^{T} \|\mathbf{u}_i^{(j)}\|_2^2 \right)$$

Here $\mathbf{U}^{(j)}$ is the orthonormal column basis of $\hat{\mathbf{Z}}^{(j)}$ for every $j \in [T]$. $\qquad\square$

## C.2. Proof of Lemma 6.3

**Lemma C.2.** *For the given regularization parameter $\lambda$, the total sensitivity scores or the sum of the sensitivity scores in the VFL setup with $[T]$ clients are upper bounded by $O\left(T \cdot \sum_{j=1}^{T} sd(\mathbf{Z}^{(j)}, \lambda, 2)\right)$.*

*Proof.* This proof is similar to the proof of lemma B.3.

Consider singular value decomposition of $\hat{\mathbf{Z}}^{(j)}$, i.e., $\hat{\mathbf{Z}}^{(j)} = \mathbf{U}\Sigma\mathbf{V}^\top$, where $\mathbf{U} \in \mathbb{R}^{n \times d_j}$ representing the orthonormal column basis of $\hat{\mathbf{Z}}^{(j)}$, $\Sigma$ is a $d_j \times d_j$ diagonal matrix and $\mathbf{V} \in \mathbb{R}^{d_j \times d_j}$ orthonomal row basis of $\hat{\mathbf{Z}}^{(j)}$. Let $\hat{\Sigma} = (\Sigma^2 + \lambda\mathbf{I}_{d_j})^{-\frac{1}{2}}$. Let $\mathbf{M} = \begin{pmatrix} \mathbf{U}\Sigma\hat{\Sigma} \\ \mathbf{V}\sqrt{\lambda}\hat{\Sigma} \end{pmatrix}$. Notice, that $\mathbf{M}^\top \mathbf{M} = \mathbf{I}_{d_j}$ and $\hat{\mathbf{Z}}^{(j)} = \mathbf{M}\hat{\Sigma}^{-1}\mathbf{V}^\top$. Hence, $\mathbf{M}$ is the orthonormal column basis of $\hat{\mathbf{Z}}^{(j)}$. Although there are infinitely many orthonormal column basis for any given matrix, each is a rotation of another. Hence, the row norms of each orthonormal column basis are the same. So, $\sum_{i=1}^{n} \|\mathbf{m}_i\|^2 = \|\mathbf{U}\Sigma\hat{\Sigma}\|_F^2 = \|\Sigma\hat{\Sigma}\|_F^2 = \sum_{j=1}^{d_j} \frac{1}{1 + \frac{\lambda}{\sigma_i^2}}$.

Now, due to lemma C.1 the sum of upper bounds are $\sum_{i=1}^{n} g_i^{(j)} = T \cdot sd(\mathbf{Z}^{(j)}, \lambda, 2) = T \cdot \sum_{j=1}^{d_j} \frac{1}{1 + \frac{\lambda}{\sigma_i^2}}$. Hence, $G = T \cdot \sum_{j=1}^{T} sd(\mathbf{Z}^{(j)}, \lambda, 2)$. $\qquad\square$

Now, we bound the sampling complexity by applying the following Matrix Bernstein's inequality.

**Theorem C.3** (Matrix Bernstein (Tropp et al., 2015)). *Let $\mathbf{X}_1, \mathbf{X}_2, \ldots, \mathbf{X}_n$ are independent $d \times d$ random matrices such that $\forall i \in [n]$, $\|\|\mathbf{X}_i\|\| \leq b$ and $var(\|\mathbf{X}\|) \leq \sigma^2$ where $\mathbf{X} = \sum_{i=1}^{n} \mathbf{X}_i$, then for some $t > 0$,*

$$Pr\left(\|\|\mathbf{X}\|\| - \mathbb{E}[\|\|\mathbf{X}\|\|] \geq t\right) \leq d \cdot \exp\left(\frac{-t^2/2}{bt/2 + \sigma^2}\right)$$

## C.3. Proof of Lemma 6.4

**Lemma C.4.** *For a given $\mathbf{Z} \in \mathbb{R}^{n \times (d+1)}$ be the augmented matrix, let $\lambda > 0$ be a scalar and $\varepsilon \in (0, 1)$. The algorithm 3 samples a set $\mathbf{S} \subseteq \mathbf{Z}$ with appropriate weights $w : \mathbf{S} \to \mathbb{R}_{>0}$. We represent the weighted set as $\mathbf{S}_w$. If the size $\mathbf{S}$ is at least $O\left(\frac{T \sum_{j=1}^{T} sd(\mathbf{Z}^{(j)}, \lambda, 2) \log(d)}{\varepsilon^2}\right)$ then the set ensures the following guarantee with at least $0.99$ probability.*

$$(1 - \varepsilon)(\mathbf{Z}^\top \mathbf{Z} + \lambda\mathbf{I}) \preceq \mathbf{S}_w^\top \mathbf{S}_w + \lambda\mathbf{I} \preceq (1 + \varepsilon)(\mathbf{Z}^\top \mathbf{Z} + \lambda\mathbf{I})$$

*Proof.* Let $\mathbf{R}$ be the random variable that holds the sampled points. So, we have the random matrix $\mathbf{R}$ as,

$$\mathbf{R} = \begin{cases} \frac{\mathbf{z}_i\mathbf{z}_i^\top}{p_i m}, & \text{with probability } p_i, \text{ if } i^{th} \text{ row of } \mathbf{Z} \text{ sampled} \end{cases}$$

Note that $\mathbb{E}[\sum_{j=1}^m \mathbf{R}_j] = \mathbf{Z}^\top \mathbf{Z}$. So, the sum of the random matrices is unbiased and is equal to the original data matrix $\mathbf{Z}^\top \mathbf{Z}$. Here $\{\mathbf{R}_1, \ldots, \mathbf{R}_m\}$ are the random variables that hold the sampled points from the algorithm. Now we bound $\|\mathbf{R}\|_2$.

$$
\begin{aligned}
\|\mathbf{R}\| &\overset{(i)}{=} \left\| \frac{\mathbf{z}_i\mathbf{z}_i^\top}{p_i m} \right\|_2 \\
&\overset{(ii)}{=} \left\| \frac{\mathbf{z}_i\mathbf{z}_i^\top G}{g_i m} \right\|_2 \\
&\overset{(iii)}{\leq} \left\| \frac{(\mathbf{z}_i\mathbf{z}_i^\top)^\dagger \mathbf{z}_i\mathbf{z}_i^\top \hat{\mathbf{Z}}^\top \hat{\mathbf{Z}} G}{m} \right\|_2 \\
&\overset{(iv)}{\leq} \left\| \frac{\hat{\mathbf{Z}}^\top \hat{\mathbf{Z}} G}{m} \right\|_2
\end{aligned}
$$

In the above, the equality $(i)$ and $(ii)$ are by definition. In the inequality $(iii)$ we use the lower bound of $g_i = \sum_{j=1}^T g_i^{(j)} \geq \frac{(\mathbf{z}_i^\top \mathbf{q})^2}{\|\hat{\mathbf{Z}}\mathbf{q}\|_2^2}$ for every $\mathbf{q}$. In the final inequality we upper bound $\mathbf{z}_i^\top)^\dagger \mathbf{z}_i\mathbf{z}_i^\top \prec \mathbf{I}_d$.

Now, we bound $var\left( \left\| \sum_{j=1}^m \mathbf{R}_j \right\|_2 \right)$ using $var[\mathbf{R}_j] \leq \mathbb{E}[\mathbf{R}_j^2]$ for every $j \in [T]$.

$$
\begin{aligned}
var\left( \left\| \sum_{j=1}^m \mathbf{R}_j \right\|_2 \right) &\leq \mathbb{E}\left[ \left\| \sum_{j=1}^m \mathbf{R}_j^2 \right\|_2 \right] \\
&= \left\| \sum_{j=1}^m \sum_{i=1}^n \frac{(\mathbf{z}_i\mathbf{z}_i^\top)^2}{p_i m^2} \right\|_2 \\
&\overset{(i)}{\leq} \left\| \sum_{j=1}^m \sum_{i=1}^n \frac{(\mathbf{z}_i\mathbf{z}_i^\top)^\dagger (\mathbf{z}_i\mathbf{z}_i^\top)^2 \hat{\mathbf{Z}}^\top \hat{\mathbf{Z}} G}{m^2} \right\|_2 \\
&\overset{(ii)}{\leq} \left\| \frac{(\hat{\mathbf{Z}}^\top \hat{\mathbf{Z}})^2 G}{m} \right\|_2
\end{aligned}
$$

In the above analysis, the inequalities $(i)$ and $(ii)$ are same as the previous analysis in $\|\mathbf{R}\|$.

Therefore by applying Matrix Bernstein Theorem C.3 we get,

$$
\begin{aligned}
Pr\left( \left| \left\| \sum_{j=1}^m \mathbf{R}_j \right\|_2 - \left\| \hat{\mathbf{Z}}^\top \hat{\mathbf{Z}} \right\|_2 \right| \geq \varepsilon \left\| \hat{\mathbf{Z}}^\top \hat{\mathbf{Z}} \right\|_2 \right) &\leq 2d \cdot \exp\left( \frac{\frac{-(\varepsilon \|\hat{\mathbf{Z}}^\top \hat{\mathbf{Z}}\|_2)^2}{2}}{\frac{\varepsilon \|\hat{\mathbf{Z}}^\top \hat{\mathbf{Z}}\|_2^2 G}{3m} + \left\| \frac{\|\hat{\mathbf{Z}}^\top \hat{\mathbf{Z}}\|_2^2 G}{m} \right\|} \right) \\
&\leq 2d \cdot \exp\left( \frac{-\varepsilon^2}{3G/m} \right)
\end{aligned}
$$

Now to ensure that the event happens with probability at least $0.99$ we need $m \geq O\left( \frac{3G \log(d)}{\varepsilon^2} \right)$. As we know from the lemma C.2 that $G = O\left( T \cdot \sum_{j=1}^T sd(\mathbf{Z}^{(j)}, \lambda, 2) \right)$ hence we have the claimed value of $m$. $\square$

# D. More Empirical Evaluations

Here, we show more rigorous experiments over more real-world datasets. Building on the setup used in the main paper, each dataset was partitioned into training and testing sets, with each experiment repeated up to 5 times, and then the median performance of these repetitions is reported.

- The Wave Energy Converters dataset consists of positions and absorbed power outputs of wave energy converters (WECs) in four real wave scenarios from the southern coast of Australia (Sydney, Adelaide, Perth, and Tasmania). We used the Sydney dataset that consisted of 71999 samples and 48 features after preprocessing. The dataset was split into training and testing subsets in a ratio of 4:1.

- The Year Prediction UCI ML dataset aims to predict the release year of a song based on audio features, specifically timbre attributes. It includes 90 attributes—12 representing timbre averages and 78 representing timbre covariances—across a range of songs from 1922 to 2011. The dataset was split into training (463,715 examples) and testing (51,630 examples) subsets, ensuring no artist appears in both sets.

- The KDD Cup dataset is used for network intrusion detection. It contains 125,973 instances of network traffic data with 122 attributes. The dataset was split into 4:1 train-test sets for intrusion detection tasks, where the goal is to classify network traffic as either normal or intrusive.

- The Credit Card Fraud Detection dataset contains 284,315 legitimate transactions and 492 fraudulent cases, resulting in a highly imbalanced dataset. We address this imbalance using SMOTE sampling, which generates a balanced dataset of 568,630 samples while maintaining the original split ratio. The task is to identify fraudulent transactions.

- Gold Price Financial Markets dataset captures 50 market indicators over 3,904 trading days. The task is to predict the future movements of stock prices, making it a time-series regression problem.

- The Blog Feedback dataset consists of 56,239 samples and 280 features and is used to predict the popularity of blog posts. The dataset contains a mix of continuous and categorical features, making it suitable for regression tasks.

- The UJIIndoorLoc UCI ML dataset is used for indoor positioning and consists of 21,048 samples with 527 WiFi signal attributes. The task is a multi-target regression problem, where the goal is to predict the latitude and longitude of a mobile device based on its WiFi signal strengths.

### D.1. VRLog Experiments

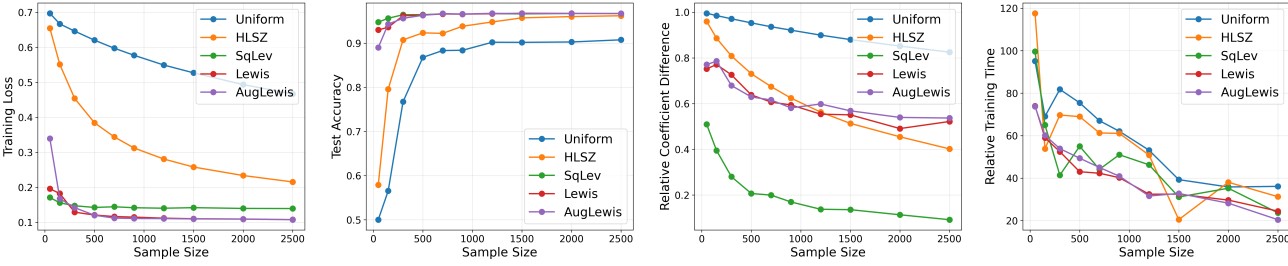

*Figure 3.* VRLog Coreset Performance (KDD Cup)

| Samples→ | 50 | | 2500 | |
|---|---|---|---|---|
| **Methods↓** | Train | Test | Train | Test |
| Uniform | 0.3419 | 0.3412 | 0.9184 | 0.9155 |
| HLSZ | 0.4717 | 0.4702 | 0.9685 | 0.9659 |
| Lewis | 0.8715 | 0.8688 | 0.9712 | 0.9684 |
| AugLewis | **0.8801** | **0.8772** | **0.9713** | **0.9685** |

*Table 2.* F1 scores on the KDD Cup dataset.

Here, we have considered KDD dataset with similar setup as we had for the credit card dataset in the main section of the paper. In figure 3 we again observe that our sampling methods outperforms the other sampling methods in all aspects, but the difference between trained model on the subsample and full dataset. Even though SqLev performs better in terms of the model difference, however there are no known theoretical guarantees in this regard for Logistic regression from

any sampling methods. Further, SqLev known to have coresets whose size is proportional to $\sqrt{n}$, makes it less reliable in practice, where $n$ is the number of data points in the training set.

We again compared the F1 scores between all the sampling methods on the KDD datasets. Similar to the Credit Card dataset, we observe that our sampling method outperforms others in the table 2, even though there are no known theoretical claims.

Now, extending our experiments from section 7, on the Credit Card Fraud Detection dataset, this time, the coresets were partitioned into five clients, where each client consisted of around 6 features. Here and we have considered various $\lambda$ values. We compared our sampling method *Augmented Coreset with Lewis* with other sampling methods, which are (1) Uniform, (2) Class-wise QR, and (3) Coreset With Lewis are SqLev and Lewis from the section 7 respectively.

We observe in Figure 4, the plots are consistent, where our coreset outperforms all the other sampling methods (Uniform, QR, Lewis Weights) in both the training and testing phases. In the QR sampling, we partition the dataset based on the labels and then compute QR decomposition on each partition separately. Next, the row norms were used to define the distribution over the training dataset, and then it was sampled. In the Lewis weights sampling, our sampling method here does not consider the regularization term in the Lewis weight approximation.

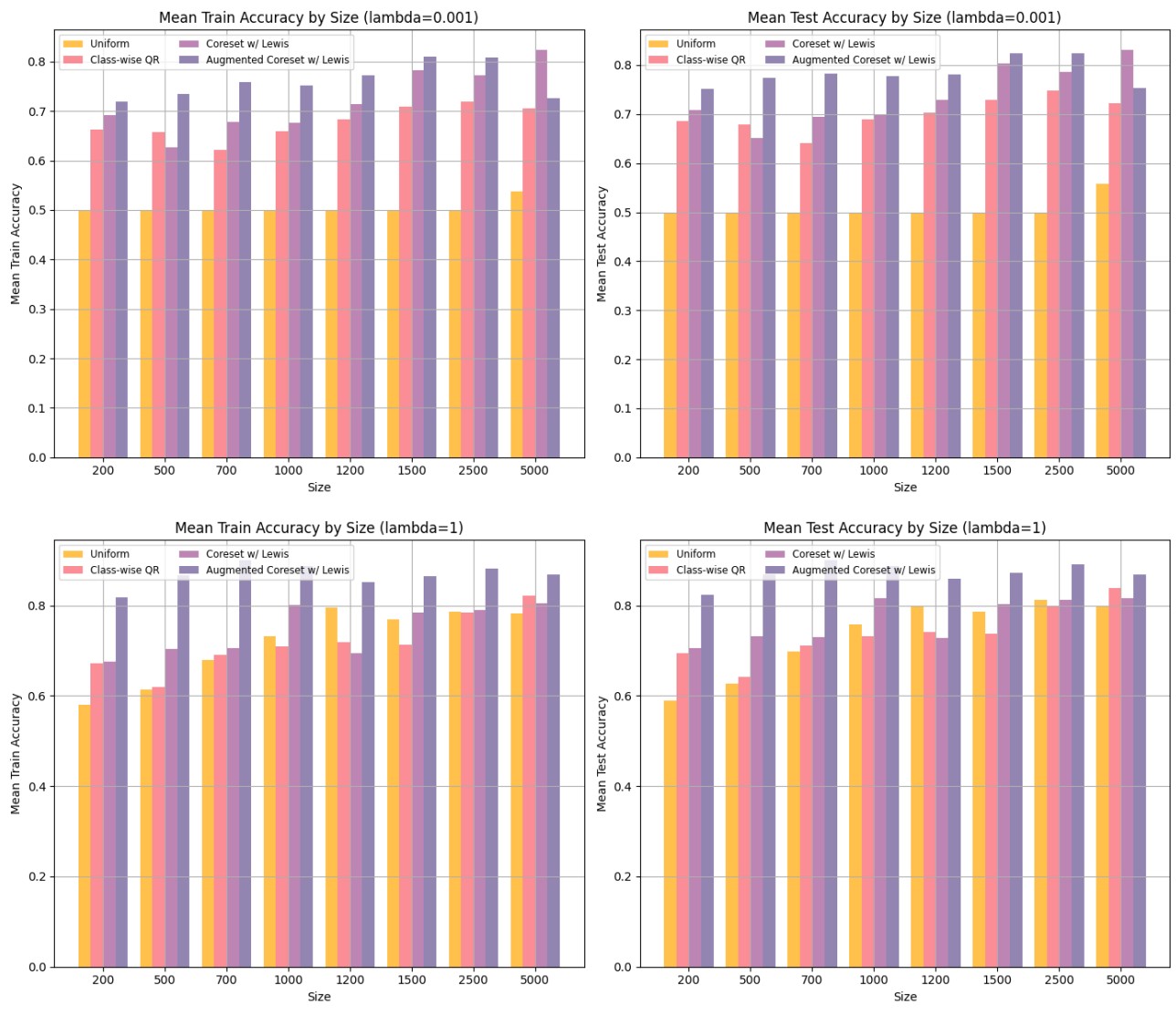

*Figure 4.* VRLog Coreset Performance (Credit Card)

## D.2. VRLR Experiments

Here, we also increase the number of clients. We have considered the regularization parameter $\lambda$ to be 102 for the Wave dataset and 104 for the Year dataset. We have considered similar competitive sampling methods as we had in the section 7. Our coreset outperforms both in the training phase as well as the testing phase in both datasets.

In the top two images of Figure 5, we show our results in the Wave Energy Converters dataset. We partitioned the coreset into 3 and 10 clients for better analysis. We have considered various regularization parameter values $\lambda$. These are captured by the parameter $\alpha$, which is defined as $\alpha = 1/\lambda$. Notice, our coreset (i.e., Ridge Leverage) performance significantly improves compared to uniform and (Huang et al., 2022), which is Leverage. Even with a coreset size as small as 0.5 percent of the full data, the performance of our coreset is significant.

Next, we tried an analysis on the Year Prediction UCI ML dataset. Here, we partitioned the coreset into 5 clients, each with around 18 features. The bottom two images in Figure 5 report the MSE of the Year dataset. While our coreset does not improve the results significantly compared to the other two sampling methods, it consistently outperforms them.

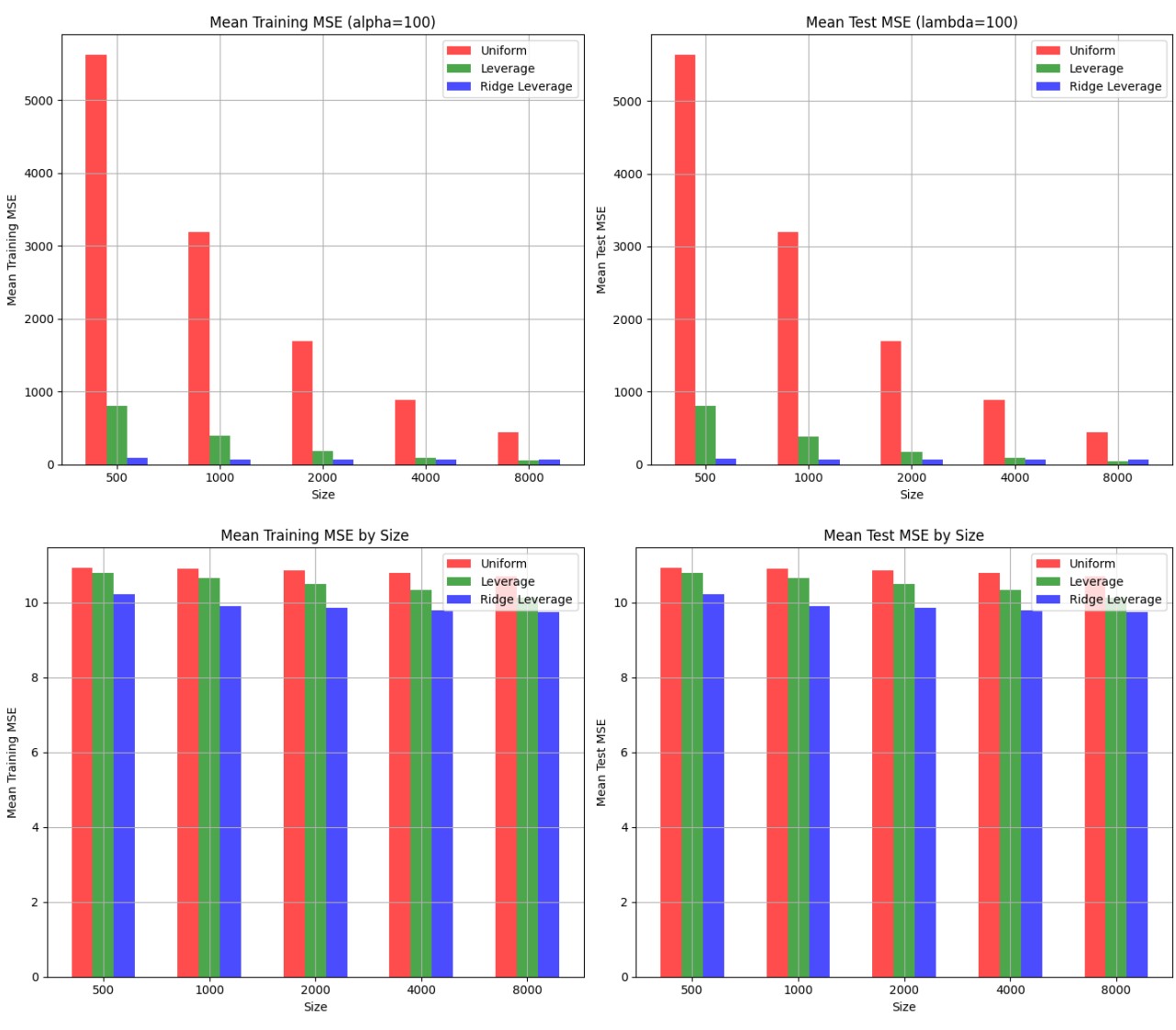

*Figure 5.* VRLR Coreset Performance (Wave Energy and Year Prediction)

We extended our experiments to additional datasets - UJIIndoorLoc, and Blog Feedback, which are presented in figure 6 and the Gold Price Finance dataset in figure 7 for different values of $\lambda$. To evaluate performance with all three sampling algorithms further, we also reported relative training time and model closeness as done in the section 7. The relative training time is defined as the ratio of the time taken to train a model on the complete training dataset to the time taken to train a model on the coreset. These are better when they are greater. The model closeness is the relative measure between the Euclidean distance between a trained model on the subsample and a trained model on the complete dataset, to the trained model from the complete dataset. These are better when smaller.

The training and the test RMSE are very similar because both train and test well represent the distribution of the population. Here, also notice that our algorithm 2 clearly outperforms all the other coreset construction methods.

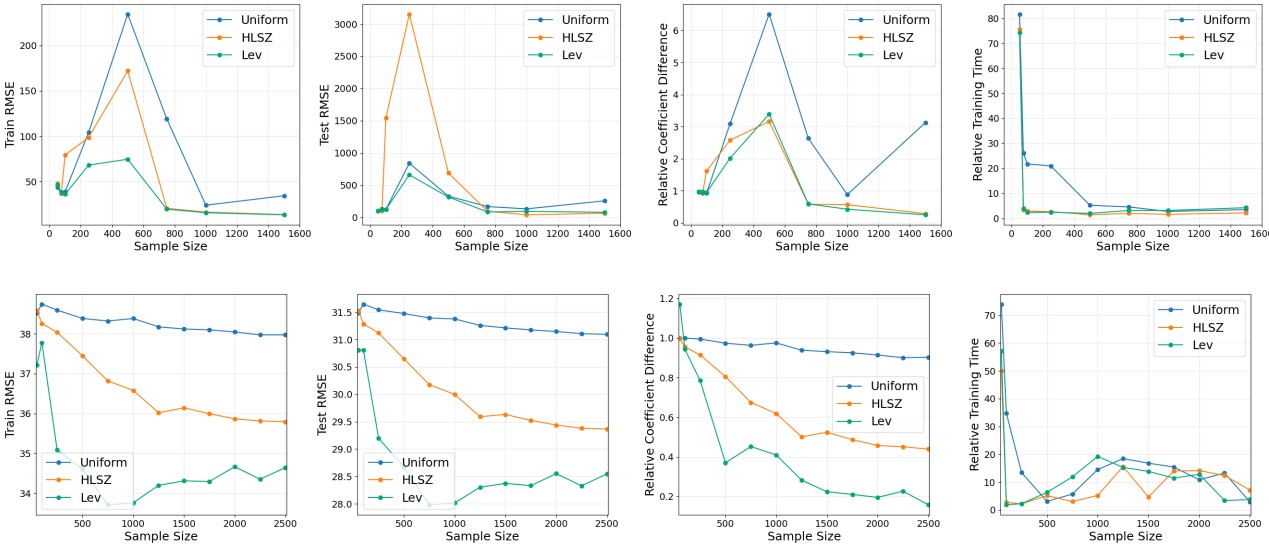

*Figure 6.* VRLR Coreset Performance (UJIIndoorLoc and Blog Feedback)

Based on this extensive empirical evidence and established theoretical guarantees, we again reiterate that our algorithm 1 and algorithm 2, which leverages regularized sensitivity scores, offers greater reliability and superiority in constructing coresets for VRLog problems and VRLR problems, respectively.

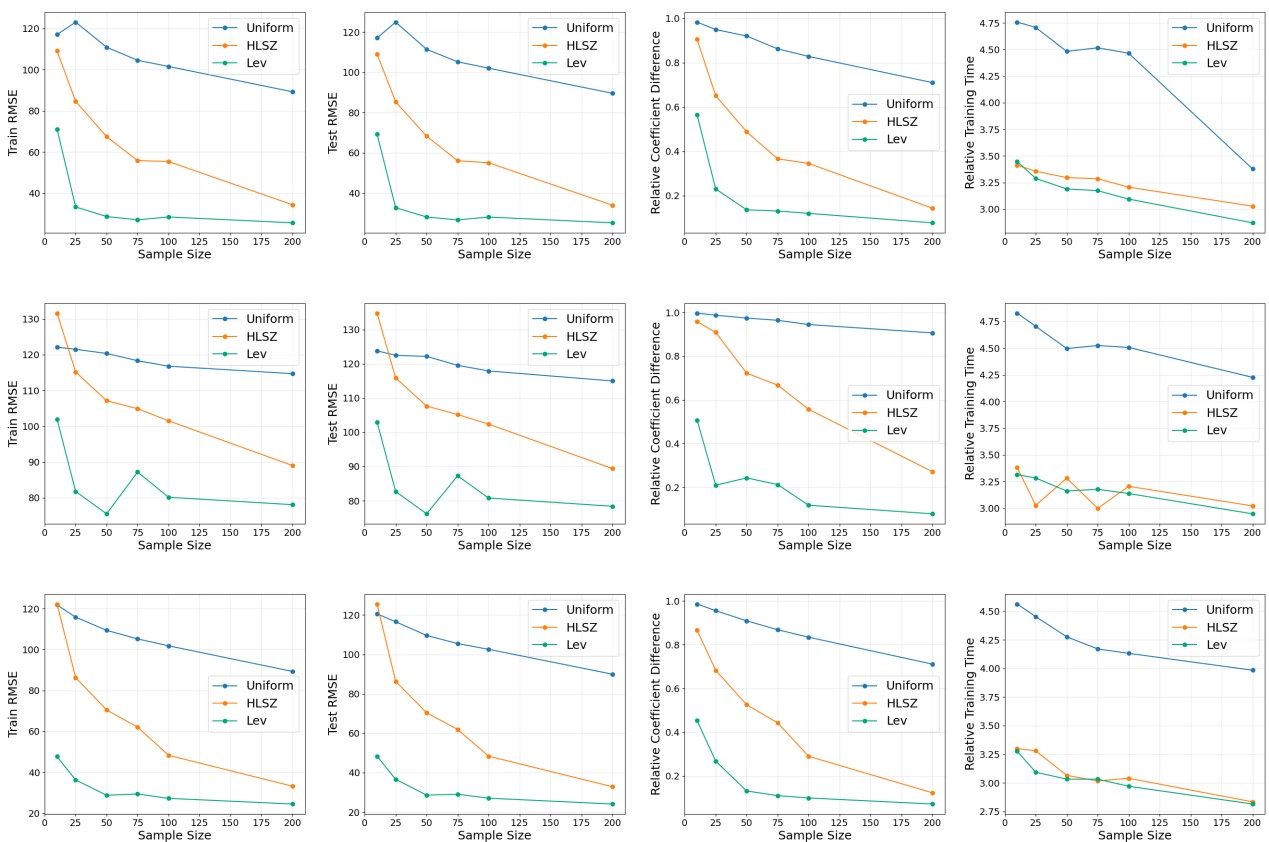

*Figure 7.* VRLR Coreset Performance (Financial)

