# OpenReview forum: "Improved Coresets for Vertical Federated Learning: Regularized Linear and Logistic Regressions"
_ICML.cc/2025/Conference — ICML 2025 poster_

### Official Review · Reviewer_YB3Z · 2025-03-13

**Overall Recommendation:** 3

**Summary:**

Coresets serve as a compact summary of training data, offering an efficient method for reducing data processing and storage complexity during training. In the context of vertical federated learning (VFL), where different clients hold distinct data features, coresets help reduce communication complexity.

This work introduces a coreset construction method for regularized logistic regression in both centralized and VFL settings. Additionally, the authors improve coreset size efficiency for regularized linear regression in VFL, eliminating its dependence on certain data properties inherent to the VFL framework. These improvements stem from novel coreset construction algorithms that account for reduced model complexity due to regularization.

Extensive empirical evaluations support the theoretical findings, demonstrating the effectiveness of the proposed coresets. The performance is further validated by comparing models trained on full datasets versus those trained on the coresets, showcasing their practical utility.

**Claims And Evidence:**

Almost all claims in the paper are well-supported by **proofs or references**, ensuring a strong theoretical foundation.

However, certain statements, while **technically correct**, may lack sufficient clarity for a **non-expert audience**. For example, the statement:
*"This is the most computationally expensive operation, which takes \(O(nd^2_j)\) time."* (Lines 355–357, right side, Page 7)

Although this statement provides the computational complexity, it does not explain **why** this operation is the most expensive or how the complexity arises from the underlying mathematical formulation. A brief explanation of the **source of this complexity**, its practical implications, and how it compares to other computational steps in the algorithm would enhance readability and understanding.

Providing additional context in such cases would make the paper more accessible to a broader audience, including those less familiar with the specific computational details.

**Essential References Not Discussed:**

The paper discusses federated learning methods designed to address client heterogeneity, such as SCAFFOLD. However, it would also be valuable to mention other relevant approaches, including ProxSkip, FedLin, and DANE, which have been proposed to mitigate the effects of heterogeneity and improve convergence in federated learning settings. A more comprehensive discussion of these methods could provide a broader perspective on existing solutions and how they relate to the proposed approach.

Mishchenko, Konstantin, et al. "Proxskip: Yes! local gradient steps provably lead to communication acceleration! finally!." International Conference on Machine Learning. PMLR, 2022.

Mitra, Aritra, et al. "Linear convergence in federated learning: Tackling client heterogeneity and sparse gradients." Advances in Neural Information Processing Systems 34 (2021): 14606-14619.

Jiang, Xiaowen, Anton Rodomanov, and Sebastian U. Stich. "Federated Optimization with Doubly Regularized Drift Correction." International Conference on Machine Learning. PMLR, 2024.

**Experimental Designs Or Analyses:**

The authors conducted experiments on both regularized logistic regression (VRLog) and regularized linear regression (VRLR) using three datasets: Credit Card (classification), Financial (regression), and Blog Feedback (regression). Each dataset was partitioned into training and testing sets, with the training data further distributed among three clients.

Maintaining the VFL sampling technique from Algorithm 1, the authors compared their sampling method with various other techniques. Once a sample was drawn using one of the sampling methods, they trained a model with a regularization parameter. For VRLog, they evaluated training loss, test accuracy, model closeness, and training time. For VRLR, they reported test RMSE and model closeness.

For each sampling method and sample size, the authors repeated the experiment 10 times and reported the median values of the results.

While I appreciate the experimental results presented on these datasets, it would be beneficial to explore additional datasets to further strengthen the evaluation and generalizability of the proposed methods. A more diverse selection of datasets could provide deeper insights into the effectiveness and limitations of the approach across different data distributions and problem settings.

The inclusion of logistic regression and linear regression experiments is particularly valuable, as they align well with the theoretical framework and provide a solid foundation for validating the theoretical findings. However, extending the experiments to include deep learning tasks could offer additional practical insights. Given the increasing importance of deep learning in real-world applications, evaluating the proposed methods on more complex models could help assess their scalability, robustness, and applicability beyond the theoretical setting.

**Methods And Evaluation Criteria:**

In this paper, the authors provide theoretical guarantees alongside experimental results for regularized logistic and linear regression. This approach is well-justified, as it ensures alignment between theoretical insights and practical validation.

**Other Comments Or Suggestions:**

Please review and address the issues raised in the previous sections.

**Other Strengths And Weaknesses:**

The paper is generally well-written and easy to follow. However, for non-experts, the theoretical section may be challenging to comprehend, even at a high level. Providing additional explanations or intuitive insights could enhance accessibility for a broader audience.

**Questions For Authors:**

Would it be possible to empirically evaluate the proposed methods on deep learning tasks to assess their practical effectiveness?

**Relation To Broader Scientific Literature:**

This work is related to both vertical federated learning and the broader federated learning literature. However, as I am not an expert in this specific area, I am unable to provide a more detailed assessment of the underlying ideas.

**Theoretical Claims:**

All theoretical claims are well-formulated, and the definitions, lemmas, and theorems are clearly stated. Additionally, the other theoretical derivations are presented in a clear manner. Unfortunately, as I am not an expert in this field, I cannot verify the proofs. However, at a high level, the statements appear to be sound and make sense.

---

> ### Author Rebuttal · Authors · 2025-03-28
>
> We thank the reviewer for taking time to read our submission and providing useful remarks. Below we address your concerns
>
> > However, certain statements, while technically correct, may lack sufficient clarity for a non-expert audience. For example, the statement: "This is the most computationally expensive operation, which takes $(O(nd^2_j))$ time." (Lines 355–357, right side, Page 7). Although this statement provides the computational complexity, it does not explain why this operation is the most expensive or how the complexity arises from the underlying mathematical formulation. A brief explanation of the source of this complexity, its practical implications, and how it compares to other computational steps in the algorithm would enhance readability and understanding.
>
> The paper mentions that, the running time of computing $g^{(j)}$ of line 6 in Algorithm 4 $O(nd_j^{2})$ for client $j \in [T]$ having the dataset $Z^{(j)}$. This is because, for $p=2$, which is the case for VRLR, the matrix $W$ is always identity $I_{n}$ and hence $g^{(j)}$ can be computed using the orthonormal column basis of $\hat{Z}^{(j)}$, since $g_i^{(j)} = (x_i^{(j)})^{T}((X^{(j)})^{T}X^{(j)}+\lambda I_{d_j})^{-1}(x_i^{(j)})= (x_i^{(j)})^{T}((\hat{Z}^{(j)})^{T}\hat{Z}^{(j)})^{-1}(x_i^{(j)})= ||u_i^{(j)}||^2$, where $u_i^{(j)}$ is the $i^{th}$ row of the orthonormal column basis of $\hat{Z}^{(j)}$. Recall, that in Huanf et. al., 22 the score for the same point is $(x_i^{(j)})^{T}((X^{(j)})^{T}X^{(j)})^{-1}(x_i^{(j)})$ which is greater than our scores because $((X^{(j)})^{T}X^{(j)}) \prec ((X^{(j)})^{T}X^{(j)}+\lambda I_{d_j})$. Hence the total sensivity is smaller and thereby our coreset size is smaller than Huang et. al. 2022 for the VRLR problem.
>
> One can compute SVD of $\hat{Z}^{(j)}$ to compute its orthonormal column basis. This can be computed in $nd_{j}^{2}$ time. Notice that these scores are part of the input to the coreset construction algorithm 1 which takes less time that $nd_{j}^{2}$. Once it has been computed, the rest of the algorithm takes fewer than $nd_{j}^{2}$ time. The first line takes  $O(n)$ to compute the sum of sensitivity scores. Next, line 2 takes $O(T)$ time at the server. In line 3 every client selects $\lceil m/T\rceil$ indices from $[n]$ based on the defined sampling probability. Next, these selected indices were shared with the server which defines global weight for the selected indices which again takes $O(n)$ time.
>
> > While I appreciate the experimental results presented on these datasets, it would be beneficial to explore additional datasets to further strengthen the evaluation and generalizability of the proposed methods. A more diverse selection of datasets could provide deeper insights into the effectiveness and limitations of the approach across different data distributions and problem settings.
>
> Thanks for your feedback. The guarantees that we provide here, as pointed out above, provably reduces the coreset size. The datasets considered are standard for this setting.
>
> > However, extending the experiments to include deep learning tasks could offer additional practical insights. Given the increasing importance of deep learning in real-world applications, evaluating the proposed methods on more complex models could help assess their scalability, robustness, and applicability beyond the theoretical setting.
>
> Since our theoretical guarantees are only specific to regularized linear and logistic regression hence, the experimental evaluations were only conducted in that domain. We did not conduct any experiments on any deeper architecture as constructing the coresets for a regularized objective would require a potentially different approach.
>
> > The paper discusses federated learning methods designed to address client heterogeneity, such as SCAFFOLD. However, it would also be valuable to mention other relevant approaches, including ProxSkip, FedLin, and DANE, which have been proposed to mitigate ... A more comprehensive discussion of these methods could provide a broader perspective on existing solutions and how they relate to the proposed approach.
>
> Indeed the literature on federated learning is now mature with methods to mitigate heterogeneity in horizontal FL (the ones mentioned by you) to federated prototype learning (FedProto by Tan et al. AAAI 22), to federated continual learning (Wang et al. CVPR 2024) to so many other settings and problems. It is almost impractical to include them in the related work of any non-survey paper. We just touched upon the aspect of heterogeneity that lies at the core of Federated ML. We included the most relevant related works given our intent and purpose of reducing the communication complexity in Vertical FL via an improved coreset construction method. We will further include more references in the final version, where additional page space will be available.
>
> We thank you again for your time and we will be happy to provide further clarifications if any.

---

> > ### Comment · Reviewer_YB3Z · 2025-04-02
> >
> > Thank you to the authors for their responses!
> >
> > I appreciate the clarifications and look forward to the promised adjustments being incorporated. Given the quality of the work, I will maintain my initial positive score.
> >
> > Best regards,
> > Reviewer

---

> > > ### Author Response · Authors · 2025-04-03
> > >
> > > Thank you for your acknowledgment. We will definitely include the promised texts in the CR version.
> > >
> > > Best regards,
> > >  Authors of the submission.

---

### Official Review · Reviewer_VbiJ · 2025-03-14

**Overall Recommendation:** 4

**Summary:**

The paper introduces a coreset construction algorithms for Vertical Federated Learning (VFL), focusing on regularized logistic and linear regression (ridge regression). The authors present algorithms to efficiently construct coresets that significantly reduce communication complexity in VFL, essential due to clients possessing different subsets of the feature space. Their primary contributions include:
1. A novel algorithm for regularized logistic regression coreset construction in centralized and VFL settings, employing ℓ₁ Lewis weights.
Improved coreset construction for vertical regularized linear regression (ridge regression), reducing coreset sizes while eliminating dependency on a specific data-related property previously considered necessary.

2. A detailed analysis demonstrating that increasing regularization parameters reduces model complexity and consequently reduces the required coreset size.

3. Empirical validation that demonstrates superior performance of their coresets compared to existing methods, both in terms of accuracy and computational efficiency.

The experiments highlight significant speed-ups (up to 100x) and comparable accuracy to models trained on complete datasets, validating their theoretical claims.

**Claims And Evidence:**

The paper clearly demonstrates:
1. The theoretical guarantees of coresets constructed with their proposed algorithms.
2. Reduction in coreset size directly correlated with increasing regularization parameter λ.
3. Empirical evidence across multiple datasets validates the theoretical results and effectiveness compared to competing methods.

**Essential References Not Discussed:**

The paper has comprehensively cited the core relevant literature and key foundational works.

**Experimental Designs Or Analyses:**

Authors validate their coresets across multiple datasets (Credit Card, Financial, Blog Feedback), comparing their performance on key metrics (training loss, test accuracy, RMSE, model closeness, training time) against other baseline methods.

**Methods And Evaluation Criteria:**

The proposed methods (coreset construction algorithms using Lewis weights and sensitivity scores) are well-suited to the stated problems (regularized logistic and linear regression) within the VFL framework. The evaluation criteria, including comparison against multiple benchmarks (e.g., uniform sampling, leverage scores, previous state-of-the-art methods), are appropriate and standard for the problem domain.

**Other Comments Or Suggestions:**

Careful proofreading is suggested for typographical and minor grammatical mistakes in the introduction and methods sections.

**Other Strengths And Weaknesses:**

Strengths:
------------
1. Strong theoretical contributions and well-motivated problem.
2. Relevant algorithmic contributions for the vertical federated learning setup.

Weaknesses:
----------------
1. Limited experimental validation on diverse or large-scale real-world datasets.
2. A potential computational bottleneck (e.g., calculation of Lewis weights) may restrict the practicality of the proposed algorithms in certain very large-scale federated learning scenarios.

**Questions For Authors:**

1. In your experiments, how did you select the regularization parameters (λ)? Did you use standard cross-validation or a more specialized method tailored to VFL settings?

2. Why have you used accuracy instead of F1-Score?

**Relation To Broader Scientific Literature:**

This paper establishes itself within the existing body of work on coresets and vertical federated learning. notably building upon prior literature such as Huang et al. (2022) and leveraging important foundational concepts from Lewis and sensitivity sampling.

**Theoretical Claims:**

The paper provides several theoretical results, including sensitivity score bounds (Lemma 1, Lemma 7), Lewis weights properties (Theorem 2, Lemma 3, Lemma 4), and coreset size bounds (Theorems 5 and 6).

Did not thoroughly check the proofs.

---

> ### Author Rebuttal · Authors · 2025-03-28
>
> We are thankful to the reviewer for taking the time to read our submission and providng a detailed review.
>
> > Limited experimental validation on diverse or large-scale real-world datasets.
>
> In the literature, it is standard to perform experiments on publicly available datasets to support the theoretical claims of federated learning algorithms. Large-scale experiments are useful in their merit to validate the distributed (multi-node-multi-GPU) machine learning algorithms, where we need to capture system-related aspects. Moreover, for a **vertical** federated learning algorithm, an experimental validation over a dataset with 280 features (blog feedback) or, for that matter, one with 500,000 samples (year-prediction) is reasonably large enough to capture the core claims of the algorithm.
>
> > A potential computational bottleneck (e.g., calculation of Lewis weights) may restrict the practicality of the proposed algorithms in certain very large-scale federated learning scenarios.
>
> This concern may not be completely well founded as a Lewis weight can be approximated using Online Row Sampling by Cohen Peng 2015 in $\tilde{O}(nd^{2})$ where $n$ is the number of the points in $\~R^{d}$. This is only larger by a $\log (n)$ factor compared to the running time for VRLR. Due to this, in Corollary 1, we have the running time of our algorithm as $\tilde{O}(nd^{2})$. We will add a remark after this corollary to clarify this.
>
> > In your experiments, how did you select the regularization parameters ($\lambda$)? Did you use standard cross-validation or a more specialized method tailored to VFL settings?
>
> Though Cross-validation is used for selecting $\lambda$ in centralized settings, it is not a standard approach for hyperparameter selection in federated learning, where it is unclear where -- on a client or the server -- to cross-validate. Please note that it is actually a critical decision problem for a client to even accept a federated model, which it would have contributed to the training of. We did a general grid search for $\lambda$. See "Federated hyperparameter tuning: Challenges, baselines, and connections to weight-sharing, Khodak et al. NeurIPS 2021". We will specify this in the final version.
>
> > Why have you used accuracy instead of F1-Score?
>
> In our experiments, we showcased losses (training/tests) because our theorem ensured theoretical guarantees on these parameters. Apart from these, we have compared the performance in terms of balanced accuracy, model parameters, and improvement in the training time. Following your suggestion, we did perform experiments to compare the performance on F1-Score. Even on that metric, the proposed algorithm outperforms the competitors. See the table below. We will include more extensive results in the supplementary material in the camera ready version of the paper.
>
> ## 1. Credit Card Dataset
>
> ### Sample Size: 500
>
> | Method   | Train F1 | Test F1 |
> | -------- | -------- | ------- |
> | Uniform  | 0.8192   | 0.8185  |
> | HLSZ     | 0.8704   | 0.8712  |
> | Lewis    | 0.9220   | 0.9230  |
> | AugLewis | **0.9330**   | **0.9343**  |
>
> ### Sample Size: 2500
>
> | Method   | Train F1 | Test F1 |
> | -------- | -------- | ------- |
> | Uniform  | 0.8723   | 0.8731  |
> | HLSZ     | 0.9071   | 0.9078  |
> | Lewis    | 0.9304   | 0.9315  |
> | AugLewis | **0.9319**   | **0.9331**  |
>
> ## 2. KDD CUP Dataset
>
> ### Sample Size: 50
>
> | Method   | Train F1 | Test F1 |
> | -------- | -------- | ------- |
> | Uniform  | 0.3419   | 0.3412  |
> | HLSZ     | 0.4717   | 0.4702  |
> | Lewis    | 0.8715   | 0.8688  |
> | AugLewis | **0.8801**   | **0.8772**  |
>
>
> ### Sample Size: 2500
>
> | Method   | Train F1 | Test F1 |
> | -------- | -------- | ------- |
> | Uniform  | 0.9184   | 0.9155  |
> | HLSZ     | 0.9685   | 0.9659  |
> | Lewis    | 0.9712   | 0.9684  |
> | AugLewis | **0.9713**   | **0.9685**  |
>
>
> We thank you again for your time and we will be happy to provide further clarifications if any.

---

### Official Review · Reviewer_Br26 · 2025-03-23

**Overall Recommendation:** 2

**Summary:**

This paper studies regularized linear regression and regularized logistic regressions in the vertical federated learning (VFL) setting, where clients store different data features. The goal is to reduce communication complexity. The paper introduces coreset algorithms for these two problems and achieves improved coreset size.

**Claims And Evidence:**

No

For example, this paper claims that their corset size improves upon that of [Huang et al., 2022] for ridge linear regression. However, they do not provide an explicit comparison between their coreset size provided in Theorem 6 and that of [Huang et al., 2022]. It is unclear why their coreset size is always smaller.

**Essential References Not Discussed:**

Yes

The paper heavily uses levis weights to compute the sensitivities, which have been studied extensively. However, they do not introduce or compare with the use of levis weights in the literature clearly. I list some potential papers below:

- William B. Johnson and Gideon Schechtman. Finite dimensional subspaces of $\ell_p$. 2001.
- Varadarajan, Kasturi R. and Xin Xiao. On the Sensitivity of Shape Fitting Problems. 2012.
- Jambulapati, Arun, James R. Lee, Y. Liu and Aaron Sidford. Sparsifying Sums of Norms. 2023.

**Ethics Expertise Needed:**

["Other expertise"]

**Experimental Designs Or Analyses:**

Yes

**Methods And Evaluation Criteria:**

Yes

**Other Comments Or Suggestions:**

- It is strange to heavily strengthen that the bound for the total sensitivity is tight. If the coreset size is tight, it is interesting. For the sensitivity, a remark may be enough.

**Other Strengths And Weaknesses:**

Weaknesses:

The writing is not well structured.
- The introduction section introduces several dense math notations but does not provide the motivations for the problem.
- The novelty of this paper compared to the literature has not been clearly illustrated. For instance, Algorithm 1 looks like Algorithm 1 in [Huang et al., 2022], while there is no discussion of the difference.

**Questions For Authors:**

- Can you provide a concrete dataset example and compare the explicit corset sizes between [Huang et al., 2022] and your result? What is the exact size improvement?

**Relation To Broader Scientific Literature:**

Yes, it relates to the broader area of federated learning and data compression.

**Theoretical Claims:**

Yes.

Check the proof of Lemma 1, which is correct.

---

> ### Author Rebuttal · Authors · 2025-03-28
>
> We thank reviewer Br26 for the time taken to read our submission and the provided feedback. We address the concerns.
>
> > The introduction section introduces several dense math notations but does not provide the motivations for the problem.
>
> Regularized linear regression and regularized logistic regression are standard basic problems in the ML community. Federated learning is now well recognized for its ability to address data privacy, security, and scalability. We stated the motivation of the problem in lines 51-54 (right); we will highlight that in the final version where an additional page will be available.
>
> >The novelty of this paper compared to the literature has not been clearly illustrated. For instance, Algo 1 looks like Algo 1 in [Huang et al., 2022], while there is no discussion of the difference.
>
> In [Huag et. al., 22] the coreset size for VRLR is a function of the rank of the input data. In comparison, our coreset size depends on the statistical dimension of the data, which is strictly smaller than the rank for any regularization parameter $\lambda>0$. Algo 1 in the paper is only for completeness and has not been included in the contribution claims. The main novelties of the paper have been highlighted in the contribution claims. Algos 2 and 4 are our contributions, whereas algo 3 has been included for completeness and stated so. We discuss the improvement of coreset size in VRLR in rebuttal to reviewer YB3Z.
>
> >It is unclear why their coreset size is always smaller.
>
> > Can you provide a concrete dataset example and compare the explicit corset sizes between [Huang et al., 2022] and your result? What is the exact size improvement?
>
> The coreset size depends on factors such as the total sensitivity, approximation error $\epsilon$, failure probability and the pseudo dimension of the problem. The most standard experiment evaluations are that for a fixed coreset size, what is the $\epsilon$ or loss or accuracy, and they tend to improve as we increase the coreset size. Comparing exact sizes between various sampling methods with a fixed one of the above parameters is impractical. Notice that in plot 2, for fixed coreset size, our algorithm has a smaller test and train RMSE and also smaller $\epsilon$ compared to the sampling methods from Huang et. al. This clearly shows that with a fixed sample size, our coresets (LEV) are performing better than the existing methods. We have further conducted more experiments on other real datasets showing similar improvements included the appendix.
>
> The exact difference between the coreset size can only be described theoretically. Let us exemplify this. For simplicity assume the number of clients to be 1, which can be easily extended to a setup with multiple clients. Let $A$ be a dataset with all $n$ points in $\mathbb{R}^{d}$ such that $n/d = c$ where $c$ is a positive integer. Again, for simplicity, take its response vector $b$ to be the zero vector. Let $A = \begin{bmatrix} I \\\ \vdots \\\ I \end{bmatrix}$ where $I$ is just identity matrix in $\mathbb{R}^{d}$. In Huang et al., the sensitivity score for every point is $1/c$. Hence, the total sensitivity for $n$ points is $n/c = d$. Notice that it is irrespective of the fact if $\lambda$ is 0 or a positive scalar. However, in section 4 [line 185 right] we have clearly motivated why a smaller coreset size is expected for the case when $\lambda > 0$. So, in such a case, our sensitivity scores are $1/(c+\lambda)$. Hence, the total sensitivity score is $n/(c+\lambda) < n/c = d$. In fact, for higher values of $\lambda$, the total sensitivity score could be significantly smaller. So, theoretically, the improvement in the coreset size is at least by a factor of $c/(c+\lambda)$. We will further underscore this example in the final version.
>
> > It is strange to heavily strengthen that the bound for the total sensitivity is tight. If the coreset size is tight, it is interesting. For the sensitivity, a remark may be enough.
>
> The coreset size is proportional to the total sensitivity (also mentioned above) which is the summation of individual sensitivity scores. Having a tighter sensitivity score gives a tighter total sensitivity, further resulting in a coreset with a tighter size. We will add a remark for this.
>
> > The paper heavily uses levis weights to compute the sensitivities, which have been studied extensively. However, they do not introduce or compare with the use of levis weights in the literature clearly.
>
> While there are various methods for computing Lewis weights, however, for our purpose, it suffices to compute the Lewis weight in any of the known methods. We have used the Lewis weights computation from Cohen Peng Lp Row Sampling by Lewis Weight 2015. We have presented this as algo 3 for completeness. We appreciate the suggestions, we will include the appropriate citations in the camera ready version.
>
> Hope we have addressed your concerns and we will be happy to provide futher clarifications if needed.

---

> > ### Comment · Reviewer_Br26 · 2025-04-02
> >
> > Thanks for the response. It addresses my concern about the size comparison with Huang et al. 2022. I increased my score.
> >
> > However, I still don't like the writing style of the paper, which is summarized below.
> > - Though the authors claim that there is a short paragraph for motivation, it is still unclear. There is no reference supporting the importance of VFL or VFL coreset.
> > - Though the main contributions do not include Algorithm 1 and the way of use for Lewis weight method. It is important to clarify that these two are not new and are from the literature. A detailed discussion is necessary to avoid overclaiming the contribution.
> >
> > Given that, I think the paper can be improved significantly in another round.

---

> > > ### Author Response · Authors · 2025-04-03
> > >
> > > We thank reviewer Br26 for acknowledging our rebuttal and positively changing the score. Below, we further address the concerns raised.
> > > >Though the authors claim that there is a short paragraph for motivation, it is still unclear.
> > >
> > > We are happy to talk about this more. As stated previously, the motivation for the problem was explicitly made clear, not only in a short paragraph: "lines 51-54 (right)," but even in an extended and natural way throughout the paper, as shown below.
> > > * At the beginning of the abstract, we wrote that **Coreset, as a summary of training data, offers an
> > > efficient approach for reducing data processing and storage complexity during training. In the emerging vertical federated learning (VFL) setting, where scattered clients store different data features, it directly reduces communication complexity. In this work, we introduce coresets construction for regularized logistic regression both in centralized and VFL settings.**
> > > * We indeed consciously kept the flow as organic as possible.
> > >     - After the abstract, we formally defined the problems VRLog and VRLR in definitions 1 and 2, respectively, in the Introduction section's first paragraphs.
> > >     - Having done that, we emphasized the usefulness of coreset in lines 51-54 right and 55-56 left.
> > >     - Following this, we formally defined the required guarantees from corsets in definitions 3 and 4. By this point, we are done with discussing the usefulness of coreset for our problem.
> > >     - Next, in section 2.3 under VFL (Line 134-144 right), we explain how a model gets trained in this setup.
> > >     - Afterward, in lines 145-154 right, we emphasize that communication complexity is a core challenge for this problem, which relates to definitions 1 and 2. We formalize the results in corollary 1 and theorem 6 through coreset.
> > >     - We believe that by this point, a reader will have sufficient exposure to the problem, its background, and motivation.
> > > * More pointedly, please notice that we have precisely defined the problems to be solved with their motivation in paragraphs P1 and P2, specifying the challenges that one needs to address while constructing corsets for our problem VRLR and VRLog in VFL (see lines 169-190, right) setting.
> > >
> > > >There is no reference supporting the importance of VFL or VFL coreset.
> > >
> > > We are unsure what "reference supporting the importance of VFL or VFL corset" means. We have sufficiently discussed VFL. We cited the works, including a recent survey (Liu et al. 2024). In lines 175-198 left, we point to the literature where corsets were used in federated learning setup. The main related work to our submission is Huang et al. 2022, which is evident in the submission that we are improving on. We will be thankful to the reviewer if more relevant papers can be pointed to us.
> > >
> > > >Though the main contributions do not include Algo 1 and the way of use for Lewis weight method. It is important to clarify that these two are not new and are from the literature. A detailed discussion is necessary to avoid overclaiming the contribution.
> > >
> > > Very humbly, we disagree with the reading that we are claiming or overclaiming anything related to Lewis weight or Algo 1. We have written clearly right above Algo 3 (line 301 right) for Lewis weight that we included this for completeness. As committed earlier in our rebuttal, we will write the same for Algo 1 in the final version. We humbly state that such a statement is self-contained, and a discussion might appear redundant.
> > >
> > > >However, I still don't like the writing style of the paper
> > >
> > > In general other reviewers found the paper easy to follow. So, we regret missing pointers for a better writing style. However, it is standard to rigorously introduce the problem at the beginning, which our paper structure also aligns with. Furthermore, see these papers:
> > > - Amsel et al. Nearly optimal approximation of matrix functions by the Lanczos method. NeurIPS 24
> > > - Musco et al. Randomized block krylov methods for stronger and faster approximate singular value decomposition. NeurIPS 15
> > > - Kacham et al. Sketching algorithms and lower bounds for ridge regression. ICML 22
> > >
> > > We will consider writing an intuitive introduction before the formal description. However, this may not structurally change the paper. We will be thankful for a consensus suggestion by all the esteemed reviewers.
> > >
> > > In summary, we humbly state that the CR version of a conference paper has one extra page to expand on the submitted texts, which then includes the address of the queries of the reviewers and the commitment made by the authors, which we will also do. The submission includes the main contributions under limited space. We tried to clearly describe the core lemmas, theorems, and experiments in detail for the benefit of the readers and kept the flow of introduction to concepts natural.
> > >
> > > As there are no further queries, we humbly request the reviewer to consider reevaluating our submission in light of the above discussion during the AC-Reviewer discussion period.

---

### Decision · Program_Chairs · 2025-05-01

**Decision:**

Accept (poster)

**Comment:**

After the rebuttal, the reviewers remain concerned about the motivation, technical novelty, and comparisons with prior work. I have included some reviewer comments below:

``Introduction: What is the importance and application of vertical federated learning? Why should we consider VRlog and VRLR in vertical settings? What are their applications? What is a coreset? Why should we consider a coreset? Is coreset the only way to address the communication complexity issue? What are the prior studies/results about vertical coreset? Are there limitations in these results? Answering these questions in the introduction can make the motivation and studied problem clearer to readers who are not that familiar with vertical federated learning or coreset.

Technical novelty: Since not all algorithms are new, the contribution section should clearly mention this. Then, the technical novelty lies in adapting the prior Lewis weight to vertical coreset construction, e.g., what obstacles exist in this adaptation and how to overcome them. The current contribution section does not sufficiently distinguish the novelty of the paper and previous contributions.

Comparison with Huang et al. (2022): It is useful to provide examples to show the best theoretical improvement in corset size. The example in the author's response needs to be added properly.”

Overall, I recommend a weak accept.